# THE POLYTOPAL COMPLEX AS A FRAMEWORK TO ANALYZE MULTILAYER RELU NETWORKS

## ABSTRACT

Neural networks have shown superior performance in many different domains. However, a precise understanding of what even simple architectures actually are doing is not yet achieved, hindering the application of such architectures in safety-critical embedded systems. To improve this understanding, we think of a network as a continuous piecewise linear function. The network decomposes the input space into cells in which the network is an affine function; the resulting cells form a polytopal complex. In this paper we provide an algorithm to derive this complex. Furthermore, we capture the local and global behavior of the network by computing the maxima, minima, number of cells, local span, and curvature of the complex. With the machinery presented in this paper we can extend the validity of a neural network beyond the finite discrete test set to an open neighborhood, covering large parts of the input domain. To show the effectiveness of the proposed method we run various experiments on the effects of width, depth, and regularisation. We further find that under regularization, less cells capture more of the volume, while the total number of cells stays in the same range. Together, these findings provide novel insights into the network and its training parameters.

## 1 INTRODUCTION

**Motivation.** Consider the two circles on the left of Figure 1. To classify the data shown in figure, we trained two neural networks of size 2-50-50-1 (the number refers to the number of perceptrons in each layer), one with a bias term, and one without. Both networks achieve zero training error, yet only the right network generalizes beyond the training data by capturing the symmetry of the data. If the only available test data is close to the training data, we would not detect this problem, highlighting the need for methods which extend the validity of the network beyond the test data.

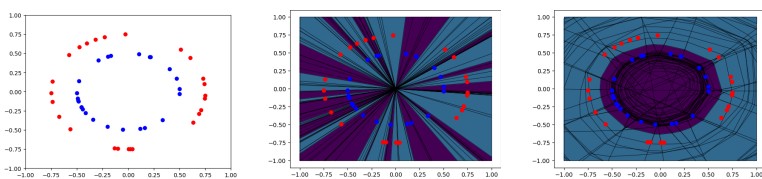

Figure 1: We trained two 2-50-50-2 MLPs, one with bias term, shown on the right, and one without a bias term shown, in the middle, to classify the concentric circles. Both MLPs fitted the data perfectly, yet we only trust the right MLP.

**This paper.** We think of the network as a function from $F\colon \mathbf{R}^D \to \mathbf{R}$ defined on some bounded domain of the input space. We further assume that the network consists of piecewise linear activation functions in the hidden layers. Under these assumptions we partition the input domain into a set of polytopes $\mathscr{C}$; see Figure 2 for an illustration of how the first layer of the network decomposes the input domain.

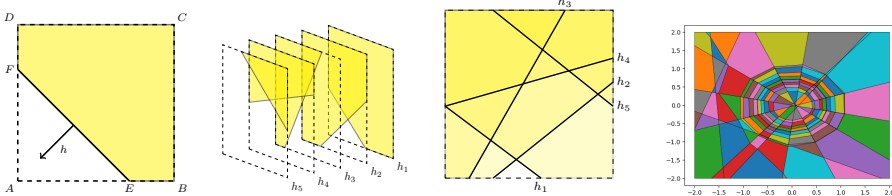

Figure 2: The **1st**-**3rd** images show respectively a single neuron separating the input domain, the neurons of the first layer doing this separately, and stacked together to give the cell decomposition after the first layer. Repeating the process for all cells gives the decomposition shown on the **right**.

Furthermore, for each vertex $v$ of the polytopal complex, we can take the star $\mathrm{st}(v)$ to capture the behavior of the network in the neighborhood of the vertex. Here the neighborhood of the vertex is the union of the cells of its star. For an arbitrary data point $x$ we take the star of the face containing $x$. We only have to consider the (finite) set of vertices of the star to analyse it and estimate the variation of $F$ around $x$. Moreover, we can assess the descent and curvature by analysing the (finite) set of faces of the star.

With the machinery of this paper we extend the validity of a neural network beyond a discrete test data point to its neighborhoods, specifically by (1) finding the cell of the complex in which $x$ lies, (2) building the star of the cells vertices, and (3) computing properties of the network on the star. Moreover, we can assess how much volume of the input domain is covered by the cells of the complex which contain test data points.

**Summary of contribution.** (1) The core of the paper is an algorithm which outputs a decomposition of the input space of a relu mlp in convex polytopes, which itself constitutes a polytopal complex. (2) We analyze the algorithm and introduce checks for the validity of the decomposition. (3) We leverage the decomposition for various analyses such as analyzing the *star* of the vertices of the complex, we capture the local behavior of the network, such as the number of extrema, the span, and the curvature. (4) We study the generalization error with our analytical framework. (5) Finally, we study the effects of depth, width, and regularization on the complex.

## 2 BACKGROUND

### 2.1 POLYTOPAL CELL COMPLEX

**Literature.** There exists a well established literature on the theoretical notions of the paper. In this section we highlight the subset used in this paper.

For pl-manifolds we refer to Rourke & Sanderson (1982). The cells of a 1-layered mlp can be understood as a hyperplane arrangement; for more on hyperplane arrangements see Stanley (2006). The theory of polytopes is covered in Ziegler (1995) and Grünbaum et al. (2003). We refer to Polthier (2002) for the definition of discrete surface curvature, for the *Griewank function* we refer to Griewank (1981), and for the *Himmelblau function* see Himmelblau (1972).

**Complex and star.** A finite family $\mathscr{C}$ of polytopes in $\mathbf{R}^D$ is called a *complex* if $(i)$ every face of a member of $\mathscr{C}$ is itself a member of $\mathscr{C}$, and $(ii)$ the intersection of any two members of $\mathscr{C}$ is a face of each of them. Let $\mathscr{C}$ be a complex and $C \in \mathscr{C}$ an element of that complex. The *star* of $C$ in $\mathscr{C}$, denoted by $\mathrm{st}(C, \mathscr{C})$, is the smallest sub-complex of $\mathscr{C}$ containing all the members of $\mathscr{C}$ which contain $C$.

### 2.1.1 ANALYSIS OF POLYTOPAL CELL COMPLEX

**Idea.** We can think of the network $F$ as a graph over its bounded domain $B \subset \mathbf{R}^D$. Since $F$ is affine when restricted to a cell $C_k$ of the network complex $\mathscr{C}$, it follows that for $x \in C_k = \mathrm{conv}\{v_1, ...v_p\}$

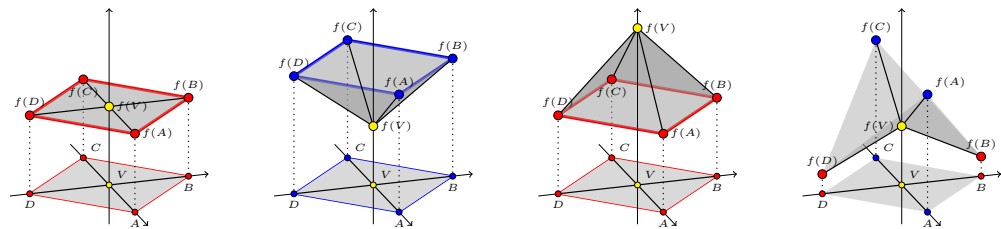

Figure 3: Examples of flat($\kappa = 0$), spherical($\kappa > 0$), and hyperbolic($\kappa < 0$) curvatures at a vertex.

the output of the network is determined by the values of $F$ at the vertices of $C_k$, i.e.

$$F(x) = F(\lambda_1 v_1 + ... + \lambda_p v_p) = \lambda_1 F(v_1) + ... + \lambda_p F(v_p), \quad \sum_i \lambda_i = 1 \tag{1}$$

**Local minima, maxima.** From Equation (1), follows that $F$ attains its minima and maxima at the vertices of the complex, with the exception of $k$-cells, at which $F$ is constant. We exclude these cases from our further discussion. We say that $F$ has a *local minimum* at vertex $v$ provided that for all vertices $w_1, ..., w_q \in \text{star}(v)$ we have $F(v) < F(w_i)$.

**Span and spectral bound.** We can further compute the global and local *span* of the star. The local span measures how much $F$ changes. Since the number of weights of the network is finite, there are only finitely many cells $C_k \in \mathscr{C}$ in the network complex. In each cell $C_k$, the network $F$ is an affine function of the type $x \mapsto \langle \nu_k, x \rangle + \rho_k$. Assuming for the moment a network without a bias term, the output of the network can be bounded by

$$||F(x)||_2^2 \leq \max_{\nu_k} |\langle \nu_k, x \rangle|^2 \leq ||x||_2^2 \max_{\nu_k} ||\nu_k||_2^2. \tag{2}$$

### 2.1.2 DISCRETE GAUSSIAN CURVATURE

**Gauss curvature.** We investigate curvature only for the two-dimensional case. Curvature measures how much we distort the graph of the network over the star compared to the flat star defined in the plane. This distortion is the difference between two terms: the sum of the interior angles of the 2-cells of the graph meeting the center $F(v)$ of the image of the star, and the sum of the interior angles of the 2-cells of the flat star meeting at the center $v$. Figure 3 shows examples of flat curvature, positive curvature (twice), and negative curvature.

**Definition.** Let $\{f_1, ..., f_m\}$ be the faces of the image of the star $\text{st}(v)$ under the network $F$, and let $\theta_i$ be the vertex angle of face $f_i$ at the vertex $F(v)$. The *Gauss curvature* $\kappa$ of a two-dimensional polyhedral surface at a vertex $v$ is defined as the vertex angle excess

$$\kappa(v) = 2\pi - \sum_{i=1}^m \theta_i(v). \tag{3}$$

### 2.2 EXPERIMENTAL SETUP

**Introduction.** We took the Himmelblau function, and the Griewank function. We sampled 6400 data points and trained MLPs on this data. The full setup is in the Appendix B.1.3.

**Himmelblau function.** The Himmelblau function has four local minima and one maximum. It has a step descent while outside of these extreme. We restricted it to the square $[-5, 5]^2$. It ranges between 0 and 890. Its local maximum is at $x = -0.3$ and $y = -0.9$, with $181.6$ and its four minima are located at $(3, 2)$, $(-2.8, 3.1)$, $(-3.8, -3.3)$, and $(3.6, -1.8)$ with value 0.

$$f(x, y) = (x^2 + y - 11)^2 + (x + y^2 - 7)^2 \tag{4}$$

**Griewank function.** We took the Griewank function restricted to the square $[-10, 10]^2$. Here we can study how the network captures the minima and maxima of the function.

$$f(x) = 1 + \frac{1}{4000} \sum_{i=1}^{2} x_i^2 - \prod_{i=1}^{2} \cos \frac{x_i}{\sqrt{i}} \tag{5}$$

# 3 COMPUTING THE DECOMPOSITION

## 3.1 ALGORITHM

**Task.** Given a multilayer perceptron $F \colon \mathbf{R}^D \to \mathbf{R}^S$ defined by its weight matrices, bias vectors, and activation functions, the *task* is to decompose a bounded box of the input space into a finite set of convex polytopes such that the network $F$ is linear on each polytope. Furthermore, we want to know the neighboring relations of the polytopes. In other words, we want to compute a *(polytopal) complex* $\mathscr{C}$ induced by the network on $B \subset \mathbf{R}^D$.

**Steps.** The proposed algorithm operates iteratively in three steps. Given a set of polytopes it partitions each polytope into sub-polytopes with the hyperplanes given by the network. For each polytope: $(\alpha)$ *It derives a partition into sub-polytopes by a set of hyperplanes.* $(\beta)$ *It identifies vertices of attached polytopes*, $(\gamma)$ *It computes the (sub)-faces of the set of polytopes.*

$(\alpha^0)$ **- Initial step.** We define the decomposition of the axis-parallel bounding box $\{b_0, ..., b_{2^D-1}\}$:

$$\text{vertices} = \{b_0, ..., b_{2^D-1}\} \tag{6}$$
$$\text{links} = \{\{b_0, b_1\}, \{b_0, b_3\}, ..., \{b_{2^D-2}, b_{2^D-1}\}\} \tag{7}$$
$$D\text{-faces} = \{\{b_0, b_1, ..., b_{2^D-1}\}\} \tag{8}$$

The intermediate steps of the algorithm only use the vertices, the links, and the $D$-faces.

$(\alpha^1)$ **- Set of cutting hyperplanes of cell.** At each cell $C_k$ the $i$-th layer of the network before the activation is an affine function. This affine function is determined by the activation pattern of the cell $C \in \mathscr{C}_{i-1}$. To compute it, we take the midpoint $\overline{x}$ of the cell, derive the activation pattern of the cell, set the corresponding rows and columns of the weights and biases to zero, and output the product $H_{i,k} = \hat{A}_i \hat{A}_{i-1} \cdots \hat{A}_0$ for the linear part and $b_{i,k} = \hat{A}_i(\hat{A}_{i-1}(...(\hat{A}_1\hat{b}_0 + \hat{b}_1) + \hat{b}_2)...) + \hat{b}_i$ for the bias term of the network at layer $i$ and cell $C$.

$(\alpha^2)$ **- Partition cell by hyperplanes.** In step $(\alpha^2)$ we iteratively go through the hyperplanes given by $(H_{i,k}, b_{i,k})$. We begin with cell $C_{k,i}$, cut it if necessary, and continue with the next hyperplane and the current decomposition of $C_{k,i}$. In this way we cut the initial cell step by step into more and more pieces.

$(\alpha^3)$ **- Partition single cell.** A cell $C$ is given by the convex hull of its vertices $\{v_0, ..., v_k\}$. The hyperplane assigns a sign to the set of vertices which we collect in $I_+$, $I_-$ and $I_0$. We further compute the cuts $J$ of the hyperplane and the cell by cutting the 1-faces of the cell. The convex hull of $J \cup I_0$ is the common face of the cuts $C_+$ and $C_-$ of the cell $C = C_+ \cup C_-$. Thus, $C_+ = \text{conv } I_+ \cup I_0 \cup J$ and $C_- = \text{conv } I_- \cup I_0 \cup J$.

$(\beta)$**-Identifying vertices.** This steps identifies vertices of polytopes which share a common face.

$(\gamma)$ **-Intersection semilattice.** Finally, we obtain the inner $k$-faces by taking the pairwise intersection of all cells and determining the affine dimension of the space spanned by the common vertices. To obtain the faces on the boundary, we intersect the cell with the 0-faces, 1-faces, 2-faces, ..., $(D-1)$-faces of the bounding box.

## 3.2 VALIDITY CHECK AND DATA STRUCTURE

**Uniqueness and independence of cutting order.** We argue that it does not matter in which order we cut the cells by the neurons of a layer. In other words we argue that the derived polytopal complex

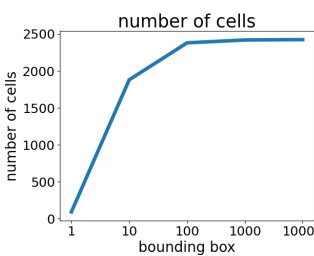 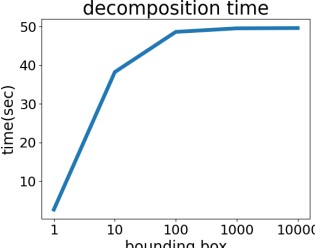

Figure 4: Computation time in seconds and number of cells of 2-30-30-1 networks trained on the Griewank data as a function of the bounding box. The plots report the mean of ten experiments.

is unique. This can be shown by induction. For a zero-layered network the algorithm outputs the polytopal complex of the bounding box which is unique. At each layer the algorithm partitions the cells by its corresponding set of hyperplanes. Reduced to each cell this partition is unique, this follows as an hyperplane arrangement uniquely partitions the input space - each partition is uniquely identified by its sign vector. It remains to show that to the partitions of two pasted adjacent cells match. This follows as the cutting (bended) hyperplanes, defined by a sign change in one neuron is continuous: Let us suppose cells $A,B$ are adjacent in layer $k$ with shared $(D-1)$-face $C$. If cell $A$ and its $(D-1)$-face $C$ is cut by hyperplane $P$ (as defined by the network in cell $A$), then there is a hyperplane $P'$ defined by the network in cell $B$ which equals $P$ when restricted to the shared $(D-1)$-face $C$. To go from $P$ to $P'$ one changes the sign vector of $P$ at the neuron of the shared $(D-1)$-face.

**Validity of decomposition.** We can assess the validity of the decomposition by (1) checking that the volume of the cells equals the volume of the bounding box, (2) we can further check if any derived vertex lies outside of the input cube, (3) checking if the sum of even faces equals the sum of odd faces plus one; in other words we compute the *Euler characteristic* of the complex, and (4) we can compare the number of 0-faces derived by computing the intersection lattice in the final step of the algorithm with the number of vertices derived during the decomposition.

**Degenerate polytopes.** Degenerate polytopes can arise during the cutting and in the final step while computing the intersection lattice. For the first case: As we cut the cells sequentially, we detect if a hyperplane leads to a degenerate polytope, as in this case all vertices of such a polytope lie in a closed half-space. For the second case: While intersecting two polytopes of the complex we derive the affine dimension of the intersection. These argument work in theory, in practice (as laid out in the appendix) almost equal hyperplanes may lead to degenerate polytopes. This is similar to meshing errors and can not be avoided in floating point or fixed-point arithmetic.

**Remark on used data structure.** Internally, we used the coordinates of the vertices $v \in \mathbf{R}^D$. For each vertex we stored its supporting hyperplanes $H_1, ..., H_k$. We further stored 0-faces, 1-faces, and $D$-faces, as list of indices, pointing to the vertices. In this way we can compute the intersection of two faces, by the set theoretic intersection of indices to determine the vertices of the intersection, followed by a computation of the rank of the intersection. Furthermore, this allows us to derive the intersection of a hyperplane and 1-face between vertex $v$ and $w$ by (1) computing the spanning vectors of its affine space - again we can achieve this by taking the intersection of indices of the supporting hyperplanes of both vertices. (2) adding the intersecting hyperplane to the supporting hyperplanes and computing its point of intersection.

### 3.3 DEPENDENCE OF BOUNDING BOX

**Number of cells and bounding box.** Increasing the bounding box of the algorithm will increase the number of cells the algorithm has to compute thus it will take longer. But above a certain value this will saturate, as no new cells are added to the bounded complex. Figure 4 shows this effect. We trained ten 2-30-30-1 networks on the Griewank data for the figure and computed the decompositions with increasing bounding boxes. Note that the training data lies in the box $[-10, 10]^2$

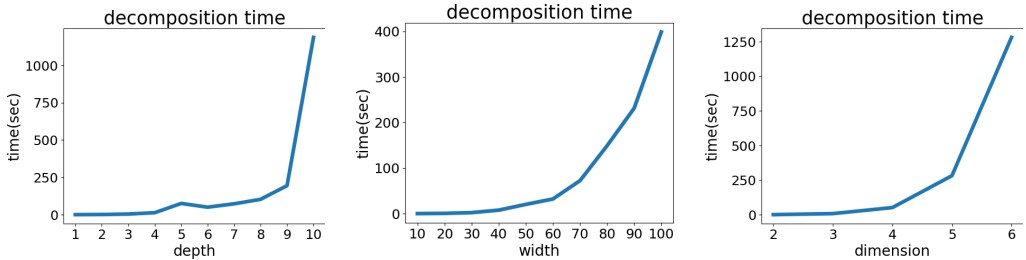

Figure 5: Computation time in seconds of the decomposition for networks of different width, depth and dimension.

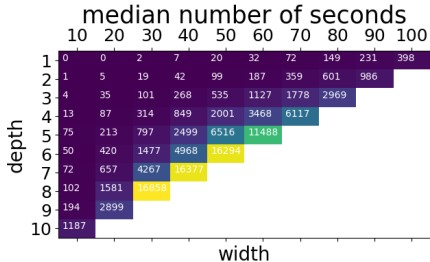

Figure 6: Computation time of the decomposition for networks of different width and depth. Each model was trained ten times.

**Remark.** We added the bounding box to the algorithm mainly to avoid numerical errors caused by floating points. If we consider networks with almost similar weights, this happens if we regularize the network heavily, then the positions of the intersecting vertices will be quite large. Combining this with vertices close to the origin is asking for numerical troubles. We can exclude this by using the bounding box. In practise this is no issue as (physical) signals of embedded systems are bounded anyway.

### 3.4 Timing

**A note on complexity.** The present algorithm and that of Berzins (2023) have some similarities. The main difference is that Berzins (2023) uses the 1-skeleton of the decomposition whereby only computing the vertices of the complex. The algorithms share enough similarity that their run time analysis carries over to our algorithm, giving a complexity of $O(|vertices|)$.

**Width, depth, and dimension.** In order to assess the run time of the algorithm empirically we trained several architectures of different width and depth on the Himmelblau data. In particular, for the width experiments we considered networks of type 2-10-1,...,2-100-1. For the depth experiments we considered networks of type 2-10-1,...,2-10-...-10-1. Furthermore, to assess dependence on the input dimension we sampled data from four concentric spheres of dimension $D$ with radii 1,2,3, and 4. The task of the network of type D-5-5-4 was to classify the correct sphere. We repeated each experiment ten times and report the median. Figure 5 shows the resulting curves. To analyze this further we varied depth and width at the same time. For the plot of Figure 6. we trained a network of type 2-*width*-...-*width*-1 on the himmelblau function, and decomposed it. We can see that deeper models tend to take longer than wider models.

## 4 Applications

### 4.1 Measuring the generalization error

**Motivation.** *How well does MSE(dev) represent MSE(true)?* The polytopal complex provides further measures besides MSE(dev) to evaluate a trained model such as *number of linear regions*, the

| measure | 201(51) | 640(160) | 2023() | 6400(1600) |
|---|---|---|---|---|
| MSE(true) | $175.18^{\pm 60.85}$ | $48.66^{\pm 13.70}$ | $15.30^{\pm 5.69}$ | $7.66^{\pm 3.88}$ |
| MSE(train) | $15.39^{\pm 7.62}$ | $18.24^{\pm 6.47}$ | $11.04^{\pm 4.43}$ | $6.69^{\pm 3.53}$ |
| MSE(dev) | $189.87^{\pm 98.01}$ | $42.57^{\pm 17.64}$ | $15.13^{\pm 6.91}$ | $7.90^{\pm 4.06}$ |
| CELLS(total) | $627.40^{\pm 174.10}$ | $734.80^{\pm 406.71}$ | $612.60^{\pm 272.80}$ | $874.00^{\pm 473.24}$ |
| CELLS(train) | $0.21^{\pm 0.05}$ | $0.38^{\pm 0.09}$ | $0.60^{\pm 0.06}$ | $0.69^{\pm 0.07}$ |
| CELLS(dev) | $0.07^{\pm 0.02}$ | $0.18^{\pm 0.06}$ | $0.37^{\pm 0.06}$ | $0.51^{\pm 0.09}$ |
| VOL(train) | $63.04^{\pm 3.95}$ | $83.53^{\pm 5.89}$ | $95.65^{\pm 2.97}$ | $98.33^{\pm 1.12}$ |
| VOL(dev) | $30.03^{\pm 3.01}$ | $53.23^{\pm 9.04}$ | $79.25^{\pm 8.29}$ | $91.03^{\pm 5.46}$ |

Table 1: For each column we trained ten 2-30-30-1 MLPs on (201+51, 640+160, 2023+506, 6400+1600) data points (train+dev) uniformly sampled from [-5,5]x[-5,5] MSE(true) was estimated with 50000 data points sampled uniformly within [-5,5]x[-5,5]. CELLS(total) reports the number of linear regions, CELLS reports the number of cells which contain train- or dev-data, VOL reports the percentage of volume of these cells.

| measure | $\sim \Sigma((0,0),1)$ | $\sim \Sigma((0,0),2)$ | $\sim \Sigma((0,0),3)$ | $\sim \Sigma((0,0),4)$ |
|---|---|---|---|---|
| MSE(true) | $11921.57^{\pm 1015.29}$ | $119.19^{\pm 93.01}$ | $71.58^{\pm 38.11}$ | $694.31^{\pm 268.91}$ |
| MSE(train) | $0.31^{\pm 0.16}$ | $31.67^{\pm 26.72}$ | $90.94^{\pm 34.12}$ | $1759.20^{\pm 752.39}$ |
| MSE(dev) | $2.42^{\pm 2.40}$ | $290.48^{\pm 327.24}$ | $10853.04^{\pm 16563.03}$ | $66372.37^{\pm 71741.04}$ |
| MSE(train-bd) | $0.31^{\pm 0.16}$ | $22.73^{\pm 18.57}$ | $41.58^{\pm 20.30}$ | $640.33^{\pm 299.30}$ |
| MSE(dev-bd) | $2.42^{\pm 2.40}$ | $27.11^{\pm 22.15}$ | $46.43^{\pm 21.69}$ | $662.40^{\pm 287.36}$ |
| CELLS(total) | $560.70^{\pm 47.03}$ | $722.70^{\pm 327.42}$ | $758.20^{\pm 176.41}$ | $400.50^{\pm 60.52}$ |
| CELLS(train) | $0.34^{\pm 0.03}$ | $0.55^{\pm 0.08}$ | $0.60^{\pm 0.05}$ | $0.63^{\pm 0.06}$ |
| CELLS(dev) | $0.23^{\pm 0.02}$ | $0.36^{\pm 0.08}$ | $0.39^{\pm 0.05}$ | $0.44^{\pm 0.06}$ |
| VOL(train) | $46.53^{\pm 2.47}$ | $92.27^{\pm 2.96}$ | $96.19^{\pm 0.93}$ | $98.34^{\pm 0.39}$ |
| VOL(dev) | $35.18^{\pm 3.00}$ | $79.80^{\pm 6.86}$ | $85.28^{\pm 2.31}$ | $92.51^{\pm 1.39}$ |

Table 2: For each column we trained ten 2-30-30-1 MLPs on (6400+1600) data points sampled from a normal distribution centered at (0,0) with increasing $\sigma$. An explanation of the rows names is given in Table 1, in addition in MSE(train-bd) and MSE(dev-bd) we only considered data points inside of [-5,5]x[-5,5].

*percentage of cells* which contain training data (and development data), the *volume* of these cells. The experiments in this section evaluate these measures. We measure MSE(true) by sampling uniformly 50000 data points form [-5,5]x[-5,5].

**Increasing data size.** For each column of Table 1 we trained ten MLPs of type 2-30-30-1 to fit the Himmelblau function with increasing data sizes. As we can see MSE(dev) follows the true MSE, whereas MSE(train) fails to capture it for smaller data sizes. We observe further that the number of total number of cells does not correlate with MSE(true), instead looking at the number of cells which actually contain data points and their volume gives a more informed answer if we can trust the model.

**Distributional shift.** For each column of Table 2 we trained ten MLPs of type 2-30-30-1 to fit the Himmelblau function with fixed data size of 6400 training points sampled from a *normal distribution* centered at the origin with increasing $\sigma \in \{1, 2, 3, 4\}$. For small $\sigma$ the data lies entirely in the bounding box, whereas for large $\sigma$, some data points lie outside of this box. Before discussing the table, we note that the Himmelblau function has a steep descent for large $||x||$, we further note that the normal distribution is symmetric, thus at the corners of [-5,5]x[-5,5] the model see less data during training for smaller values of $\sigma$. We observe that MSE(dev) does not match MSE(true). Furthermore, MSE(train-bd) and MSE(train-dev) are close to each for $\sigma \in \{2, 3, 4\}$, but they match MSE(true) only for $\sigma = 4$. We further note in the experiment that the number of cells is no proxy for good generalisation. Only if the data covers large parts of the volume does MSE(dev-bd) match MSE(true), this is caused by the distributional shift. Finally, we note that already a few cells cover large parts of the volume.

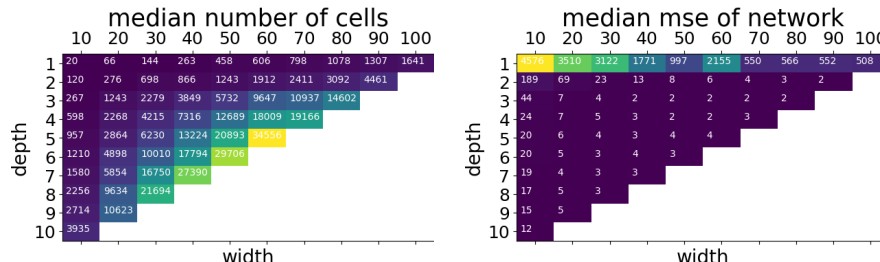

Figure 7: For each shown combination of depths and width we trained ten networks on the Himmelblau data.

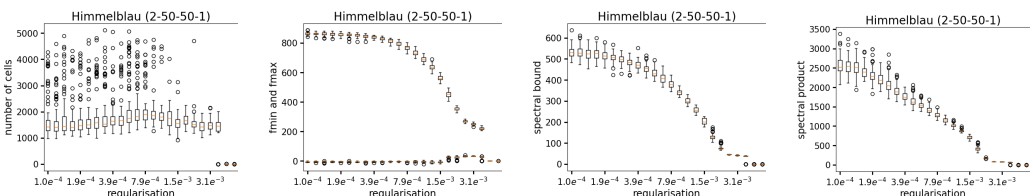

Figure 8: Each plot shows the quantiles of 24x100 trained MLPs. The plot on the left shows the number of cells, the next plot shows the spans maximal and minimal span of the stars, the third plot shows the spectral bound of the complex, the right most plot shows the product of the spectral norms of the weights. We see that the number of cells remains constant, whereas the other parameters decrease as we increase regularization.

## 4.2 CONTROLLING THE STRUCTURE

**Motivation.** It is desirable to control the pl-structure of the trained network. Yet at the same time this will be quite challenging. At present we can only do this indirectly. The most direct way is by choosing depth and width of the network. A more indirect way is through the training data. Finally, we can also regularize the network. In this section we study the effects of depth, width, and regularization.

**Controlling depth and width.** In this set of experiments we investigated the effect of depth and width on the number of cells of the complex, as well as the final MSE. We trained several networks of type 2-*width*-...-*width*-1 on the Himmelblau data. Figure 7 reports the results. We see that the networks should at least have two preferable three layers in order to achieve a good fit as measured by the MSE. Further we infer that increasing width and depth at the same time yields more complex networks, but with no effect on the MSE.

**Controlling regularization factor.** Regularization offers another way to control the structure of the network, yet it remains unclear what it controls. For the experiment we added $l_2$-regularization to the loss function. We considered 24 different regularization factors. For each regularization factor we trained one hundred times a MLP of type 2-50-50-1 on Himmelblau data. In Figure 8 we observe that the number of faces remains roughly constant, until training collapses. In the second plot of the figure, we notice that the span of the stars decreases, in other words as we regularize the stars become more flat. Finally, the last two plots show the spectral bound of the network defined on the bounding box, and the product of the spectral norm of the matrices. We see the regularization decreases the spectral norm, but we also note the large gap between the true spectral norm and its upper bound.

## 5 RELATED WORK

**Counting of linear regions and activation patterns.** This line of research connects the expressiveness of the network and its architecture, motivated by the questions such as *Is deeper better than wider?* Deriving the number of cells for neural network dates back to bounding the number of

cells in a hyperplane arrangement Stanley (2006). Montúfar et al. (2014) gave upper bounds on the number of linear regions of the network. This was followed by Hanin & Rolnick (2019b), Goujon et al. (2024), Wang (2022), Serra et al. (2018). Besides linear regions also activations patterns have been counted Hanin & Rolnick (2019a).

**Expressiveness of neural networks.**   It is impossible to compute the polytopal complex for networks of large input dimensions. But we can sneak into these networks by studying planes or paths in the input space. This is done for example in Raghu et al. (2017),Novak et al. (2018), Humayun et al. (2024b). Another approach is to consider a subset of the linear regions, see Gamba et al. (2022),

**Related algorithms.**   The closest algorithms to the present paper Balestriero & LeCun (2023) provides an algorithm which returns the exact number of cells and their activation pattern, but they do not return the vertices and their structure. In a sense this algorithm operates entirely on the h-representation of the polytopes.

Berzins (2023), provides an algorithm for the enumeration problem of neural networks. Their core idea is a sequential cutting of the 1-skeleton of the complex per layer. There is some similarity to step ($\alpha^3$) of our algorithm but our algorithm derives the entire structure of the polytopal complex not just the 1-skeleton. This cell structure is necessary to derive curvature and volume.

Finally, Humayun et al. (2024a), Humayun et al. (2023) provides an algorithm for two-dimensional input based on planar geometry. A deeper discussion of the related algorithms can be found in Appendix C.1.

**Polytopal complex.**   Polytopes (and convex analysis) have been used to analyse and improve properties of networks such as robustness Croce et al. (2019), bounds on architecture Arora et al. (2016). In another direction Liu et al. (2023) used the polytopal complex to study topological signals of manifolds in the input space from samples. More recently, tropical geometry has provides a related path to study neural networks Brandenburg et al. (2024).

**Vizualisation and explainability.**   Finally, we briefly mention other approaches to visualize and explain neural network, Zeiler & Fergus (2014) focusses on feature visualization. Olah et al. (2018) address the need to understand how neural networks predict on given data points, Ergen & Pilanci (2020) studied two-layer ReLU Networks based on formulating a convex optimization problem with infinitely many constraints. Lipton (2018) points out, important desiderata for interpretability are trust and informativeness. There is also a line of work Li et al. (2018) that analyses the loss landscape of neural nets to better understand training dynamics.

# 6 CONCLUSION

**Summary.**   We have provided an algorithm to compute the polytopal complex of a neural network. This allowed to compute several statistics of a trained network. We analysed an example network qualitatively and quantitatively, by looking at its level sets, by computing the percentage of the volume covered by data, among others. We have seen that width and depth control the number of cells of the decomposition on average. Regularization, in contrast, did not change the number of cells much, but it controlled the span of the stars and the spectral radius of the network.

**Limitations.**   This work is limited to neural networks with a finite computational budget, a small number of inputs, bounded input domain, and a continuous piece-wise linear activation function. However, this is a typical setting in embedded systems and virtual sensors.

**Future directions.**   We did not look at discrete curvature beyond two dimension. Another extension is the approximation of (bounded) continuous activation functions such as the *tanh* function by ReLU activations. It is practically impossible to analyze the polyhedral complex of a network with input dimension greater than ten. But similar to Humayun et al. (2024a) it would be interesting to intersect the input space with a low dimensional subspace and study the (reduced) polyhedral complex. Finding a more direct way of controlling the pl-structure different or finding a novel loss term is also an exciting and important research direction.

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

## A  APPENDIX / SUPPLEMENTAL MATERIAL

### A.1  NOTATIONS

**Neural network.**  The network considered in this papers are functions of the following internal structure.

$$F\colon \mathbf{R}^{D_0} \xrightarrow{A_0\cdot+b_0} \mathbf{R}^{D_1} \xrightarrow{\phi_0} \mathbf{R}^{D_1} \xrightarrow{A_1\cdot+b_1} \mathbf{R}^{D_2} \xrightarrow{\phi_1} \cdots \mathbf{R}^{D_L} \xrightarrow{A_L\cdot+b_L} \mathbf{R}^{D_{L+1}} \xrightarrow{\phi_L} \mathbf{R}^{D_{L+1}} \tag{9}$$

The algorithm of the paper also considers intermediate computations of the network. We denote by $F_i$ the network at layer $i$ after the activation function, and with $G_i$ the network at layer $i$ before the activation function. Schematically this looks as follows.

$$F_i\colon \mathbf{R}^{D_0} \xrightarrow{A_0\cdot+b_0} \mathbf{R}^{D_1} \xrightarrow{\phi_0} \mathbf{R}^{D_1} \cdots \mathbf{R}^{D_i} \xrightarrow{A_i\cdot+b_i} \mathbf{R}^{D_{i+1}} \xrightarrow{\phi_L} \mathbf{R}^{D_{i+1}} \tag{10}$$

and

$$G_i\colon \mathbf{R}^{D_0} \xrightarrow{A_0\cdot+b_0} \mathbf{R}^{D_1} \xrightarrow{\phi_0} \mathbf{R}^{D_1} \cdots \mathbf{R}^{D_i} \xrightarrow{A_i\cdot+b_i} \mathbf{R}^{D_{i+1}} \tag{11}$$

Here we used the following notation:

- $F$ - the *multilayer perceptron* $F\colon \mathbf{R}^D \to \mathbf{R}^S$ considered in the paper.
- $F_i$ - the output of the *multilayer perceptron* $F_i\colon \mathbf{R}^D \to \mathbf{R}^D_i$ at the $i$-th layer *after* the activation function.
- $G_i$ - the output of the *multilayer perceptron* $G_i\colon \mathbf{R}^D \to \mathbf{R}^D_i$ at the $i$-th layer *before* the activation function.
- $A, A_0, A_1, ...$ - the weight matrices of the mlp, with $A$ denoting any weight matrices, and $A_k$ denoting a weight matrix from the $k$-layer to the $(k+1)$-layer.
- $b, b_0, b_1, ...$ - the bias vectors of the mlp, with $b$ denoting any bias vector, and $b_k$ denoting the bias vector from the $k$-layer to the $(k+1)$-layer.
- $\phi, \phi_1, \phi_2, ....$ - activation functions of the network. In most cases this is either the *relu*, or a *linear* function.
- $L$ - the number of hidden layers of the considered network;
- $D, D_0, D_1, ...$ - the input dimension of the network, and the dimensions of the $k$-layer of the network.
- $S$ - the output dimension of the network;
- $(N_k, \rho_k) \in \left(\mathbf{R}^{S\times D}, \mathbf{R}^S\right), (\nu_k, \rho_k) \in \left(\mathbf{R}^D, \mathbf{R}\right)$ - defines the *affine mapping* given by the network in cell $C_k$, i.e. $F(x) = N_k x + \rho_k$ for all $x \in C_k$, for a vector output and $F(x) = \langle \nu_k, x \rangle + \rho_k$;
- $Q_k$ - matrix which switches at the *decision boundary* of cell $C_k$;
- $R_k$ - matrix which switches at the *level sets* of $F$;

**Index sets.**  We describe the vertices, $k$-faces, and hyperplanes of the arrangement by subsets of the integers. To each vertex, $k$-face, and hyperplane corresponds a geometric realisation. So the integer $v \in \mathbf{N}$ corresponds to a point $p \in \mathbf{R}^D$. One can think of the vertex as an index of a sequence of geometric points. Similarly a $k$-face is a subset of the natural numbers to which a subset of points of the Euclidean space corresponds. Analogously, a hyperplane $H$ is an index which points to a pair $(n, d) \in \mathbf{R}^D \times \mathbf{R}$.

- $v, v_0, v_1, ...$ - *vertices* of the arrangement, $v, v_0, ... \in \mathbf{N}$;
- $\mathscr{F}$ - the *face semilattice* of the decomposition;
- $\mathscr{F}^k$ - all *k-faces*;
- $\mathfrak{f}^k, \mathfrak{f}_0^k, \mathfrak{f}_1^k, ..., \mathfrak{g}^k, \mathfrak{g}_0^k, \mathfrak{g}_1^k, ... \in \mathscr{F}^k$ - *k-faces* of a complex;
- $\mathscr{A}$ - the *affine space semi lattice*;
- $\mathscr{A}^k$ - the *k-dimensional affine spaces*. Each space is spanned by a $k$-face of the semi-lattice.
- $\mathscr{H}$ - the set of hyperplanes of the arrangement;
- $\mathscr{H}^l$ - a set of $l$-many hyperplanes corresponding to the $(d-l)$-faces;

**Geometric realization.** In theory we could derive all computations here. A superscript if present indicates the dimension of the object. A subscript denotes a specific element.

- $(n, d), (n_0, d_0), (n_1, d_1), ...$ - pairs $(n_k, d_k) \in \mathbf{R}^D \times \mathbf{R}$ defining hyperplanes, or halfspaces;
- $\mathcal{V}$ - the set of vertices, $v \in \mathbf{R}^D$;
- $\mathcal{C}^0, \mathcal{C}^1, ...$ - the $k$-cells of the arrangement.
- $\mathcal{C}, C_1, C_2, ...$ - the cells of maximal dimensions of the arrangement.
- $\mathcal{H}$ - the set of hyperplanes;
- $\mathcal{H}^+$ - the set of positive halfspaces;
- $\mathcal{H}^-$ - the set of negative halfspaces;
- $H, H_0, H_1, ...$ - hyperplanes, i.e. $H_1 = H_1(n_1, d_1) = \{x \in \mathbf{R}^D | \langle n_1, x \rangle + d_1 = 0\}$;
- $H^+, H_0^+, H_1^+, ...$ - (closed) positive halfspaces, i.e. $H_1^+ = H_1^-(n_1, d_1) = \{x \in \mathbf{R}^D | \langle n_1, x \rangle + d_1 \geq 0\}$;
- $H^-, H_0^-, H_1^-, ...$ - (closed) negative halfspaces, i.e. $H_1^- = H_1^-(n_1, d_1) = \{x \in \mathbf{R}^D | \langle n_1, x \rangle + d_1 \leq 0\}$;

**Complex, link and star**

- $\mathscr{C}$ - (bounded) polytopal complex induced by the network.
- $B \subset \mathbf{R}^D$ - a bounding box on which the network is considered.
- $\mathscr{F}$ - the face complex;
- $\mathrm{st}(v)$ - the *star* of vertex $v$;

**Operations.** Here we list several operations which we are using throughout the paper.

- $\mathrm{conv}$ - the *convex hull* of a set of vertices;
- $\text{aff-dim}(v_1, ..., v_i)$ - the *affine dimension* of a set of vertices, i.e. the dimension of the affine space spanned by the vertices.
- $\partial$ - the *boundary* of a face;
- $\langle \cdot, \cdot \rangle$ - the *scalar product* of $\mathbf{R}^D$, i.e. $\langle x, y \rangle = \sum_k x_k y_k$ for $x = (x_1, ..., x_D)^T, y = (y_1, ..., y_D)^T \in \mathbf{R}^D$;
- $x \vee y$ - alternative way of writing the max function, i.e. $x \vee y = \max\{x, y\}$ we use it for the relu activation function: e.g. $\mathrm{relu}(x) = Ax + b \vee 0$. The symbol comes from lattice theory and is called *join*.
- $\mathrm{int}\, X$ - the *interior* of a given set $X$, i.e. the largest open set $O$ contained in the given set $O \subset X$. We use it to denote an open halfspace $\mathrm{int}\, H^+(n, d) = \{x \in \mathbf{R}^D | \langle n, x \rangle + d > 0\}$.
- $\mathbf{1}$ - a vector of 1's of the correct dimension.

**Properties.** Here we list symbols of properties of the network which we compute in the main paper.

- $\kappa(v)$ - the *(total) Gauss curvature* of a two dimensional polyhedral surface at vertex $p$.

## A.2 VARIANTS

**Idea.** This section summaries what can be cast in the *relu*-framework of the paper.

# B FURTHER APPLICATIONS

## B.1 ANALYZING A TRAINED NETWORK

**Summary.** Here we show, exemplarily, how to assess a trained network with the polytopal complex. To this end we trained a 2-30-30-1 network on 64 samples of the Himmelblau function for 20000

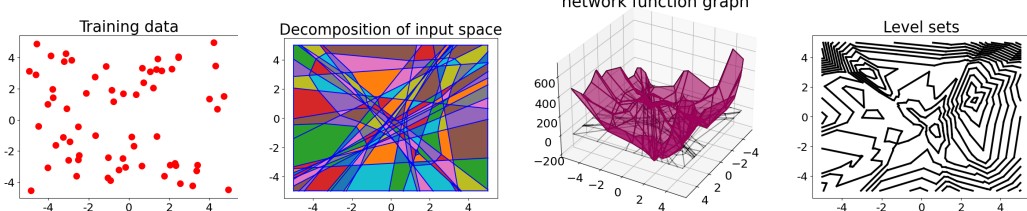

Figure 9: The figure shows the training data, the decomposition of the trained network, the network shown as a graph over its decomposition, and the level sets of a 2-30-30-1 network trained on the Himmelblau data. We point out the X-type structure of the decomposition in the second plot.

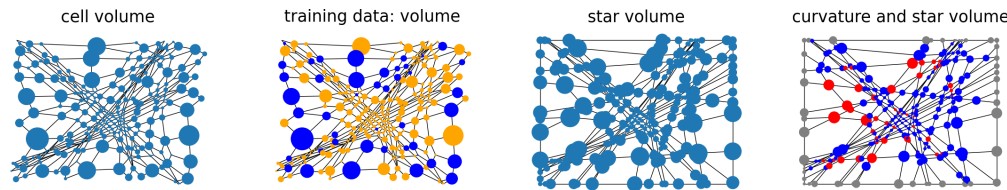

Figure 10: The first plot of the figure shows the neighborhood graph of the cells, each node scaled by its cell volume. For the second plot, we color-coded each node, with blue=contains data, and orange=no data. The third plot shows the stars of the network, again scaled by volume. The fourth plot color-codes the curvatures of the inner stars, with blue=flat and red=curved.

epochs. Figure 9 shows the training data, the decomposition of the trained network, the network shown as a graph over its decomposition, and the level sets. To compute the level sets, we encode the levels in an additional layer with weight vector $[[1, ..., 1]]$ and bias term $[l_1, l_2, ..., l_n]$. More details are given in Appendix B.1.2.

**Neighborhood graph, cell volume and training data.** The network decomposes the input box into 367 cells. Thus, we fitted the training data with 367 different linear models. If we had just one cell, and a low MSE, we certainly would trust our model. To achieve a similar level of trust for neural networks, we first compute the neighborhood graph of the cells. Here two cells are connected by an edge if they share a common face. In addition, we computed the volume of each cell, this is the volume of the convex hull of the vertices of the cell. The first plot in Figure 10 shows the neighborhood graph of the cells with the nodes scaled by the volume its cell. We observe an X-type structure clustered with small cells, and large cells outside of the X. This resembles the local extrema of the Himmelblau function quite well. The second plot of Figure 10 shows the neighborhood graph of the cells with the nodes scaled by the volume of the cells. In addition, we color-coded each node which contained training data in blue, and orange otherwise. In total 46% of the cell volume is covered by data. *This shows how to extend the discrete training data set to a neighborhood given by the union of the cells.* In our example, we see many cells without training data centered in the middle of the bounding box. Looking at the plots we would tend to mistrust the model. But this assessment may be to early as we did not use the local structure of the network. The linear models of the cells are connected! In order to assess this local structure we need the stars.

**Stars, local extrema and curvature.** The closed curves in the level set plot reveal visually that we have at least six extrema. In order to assess the model further we derived the *star* of each vertex. The third plot in Figure 10 shows the stars of the network. Again we scaled each node by the volume of its star. For the fourth plot we computed the curvature of the interior vertices and color-coded the nodes. Using the stars of the vertices we find that there are ten local minima and fourteen local maxima. Further, looking at the curvature in the center, there the linear models of the cells are not supported by data, we see that the network is mostly flat. It seems that the model is *interpolating* the data.

### B.1.1 Classification networks

**Classification layer.** In a classification layer the network $F\colon \mathbf{R}^D \to \mathbf{R}^S$ outputs the argmax-function of the vector

$$\operatorname{argmax}_k F(x) \tag{12}$$

here we *argmax* over the components of $F(x)$. At the decision boundaries of the classifier the difference of two components of $F(x)$ change sign. In other words writing $F(x) = (F_i(x), ..., F_S(x))$, for some pairs of $i, j \in \{1, ..., S\}$ we have $F_i(x) - F_j(x) = 0$. This local sign change can be encoded in a *relu* network by replacing the softmax layer of the current cell $C_k$ given by $x \mapsto N_k x + \rho_k$. Let us denote the component of this vector for $x \in C_k$ by $(\psi_1(x), ..., \psi_S(x))^T = N_k(x) + \rho_k$. Then the network outputs class $i$, if $\psi_i(x) > \psi_j(x)$ for all $j \in \{1, ..., \hat{i}, ..., S\}$. In other words $\psi_i(x) - \psi_j(x) > 0$. Now using the notation $\psi_i(x) = [N_k]_i(x) + [\rho_k]_i$. We can design a matrix $Q_k$ and bias term $q_k$ which switches sign at the decision boundary of the classifier at the current cell by setting.

$$Q_k = \begin{bmatrix} [N_k]_1 - [N_k]_2 \\ \dots \\ [N_k]_1 - [N_k]_S \\ [N_k]_2 - [N_k]_3 \\ \dots \\ [N_k]_{S-1} - [N_k]_S \end{bmatrix} \text{ and } q_k = \begin{bmatrix} [\rho_k]_1 - [\rho_k]_2 \\ \dots \\ [\rho_k]_1 - [\rho_k]_S \\ [\rho_k]_2 - [\rho_k]_3 \\ \dots \\ [\rho_k]_{S-1} - [\rho_k]_S \end{bmatrix} \tag{13}$$

**Example.** To make this concrete, suppose we wish to classify four concentric circles in the plane by a relu-network of shape $2 \to 5 \to 5 \to 4$. Suppose further we are given a cell $C_k$ with affine mapping given by matrix $N_k$ and bias term $\rho_k$. Then the input of the softmax is the four-dimensional vector $(\psi_1(x), ..., \psi_4(x))$. To catch all decision boundaries of the cell we have to take the pairwise difference of all components: $\psi_1(x) - \psi_2(x)$, $\psi_1(x) - \psi_3(x)$ and so on. With this matrix $Q_k \in \mathbf{R}^{6 \times 2}$ and bias vector $q_k \in \mathbf{R}^6$ constructed above read

$$Q_k = \begin{bmatrix} [N_k]_1 - [N_k]_2 \\ [N_k]_1 - [N_k]_3 \\ [N_k]_1 - [N_k]_4 \\ [N_k]_2 - [N_k]_3 \\ [N_k]_2 - [N_k]_4 \\ [N_k]_3 - [N_k]_4 \end{bmatrix} = \begin{bmatrix} n_{11} - n_{21} & n_{12} - n_{22} \\ n_{11} - n_{31} & n_{12} - n_{32} \\ n_{11} - n_{41} & n_{12} - n_{42} \\ n_{21} - n_{31} & n_{22} - n_{32} \\ n_{21} - n_{41} & n_{22} - n_{42} \\ n_{31} - n_{41} & n_{32} - n_{42} \end{bmatrix} \text{ and } q_k = \begin{bmatrix} [\rho_k]_1 - [\rho_k]_2 \\ [\rho_k]_1 - [\rho_k]_3 \\ [\rho_k]_1 - [\rho_k]_4 \\ [\rho_k]_2 - [\rho_k]_3 \\ [\rho_k]_2 - [\rho_k]_4 \\ [\rho_k]_3 - [\rho_k]_4 \end{bmatrix} \tag{14}$$

Here we assume that the components of $N_k$ are given by $N_k = (n_{ij})$. Each cell of this complex predicts a single class. By running through all the cells of this complex we obtain a cell decomposition by output class.

### B.1.2 Level sets

**Task.** While applying the methods described in this paper on a real setting it is desirable to now what inputs give a specific output. These subset of the input space can then be studied further to analyse if the desired behaviour of the network matches its actual behavior. Furthermore, this allows us to observe cause and effect, i.e. we can alter the weights and study how the computed subsets change increasing our trust in the networks further.

**Algorithm.** To reformulate the task, given $F\colon \mathbf{R}^D \to \mathbf{R}$ and $c \in \mathbf{R}$ we want to compute the level sets $F^{-1}(c)$. It is obvious that this level set is a pl-submanifold of the input space. To compute it note that by adding a layer of the type $x \mapsto \{x - c \wedge 0\}$ to the network we build a function which changes its sign at the level $c$. To determine the level set we simply run the algorithm to obtain a decomposition of the input space $\tilde{\mathscr{C}}$, and select all $k$-faces of $\tilde{\mathscr{C}}$ for which $F$ equals to $c$ at all vertices of the $k$-faces.

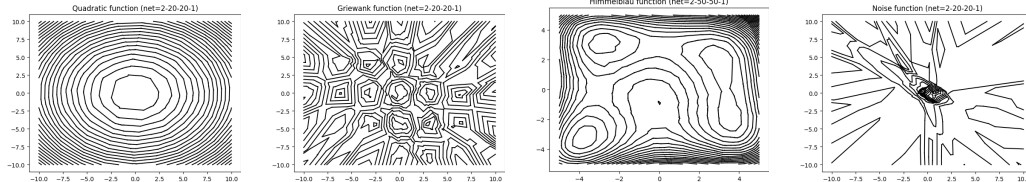

Figure 11: Level sets of networks trained on four functions considered in the paper: *quadratic function*, *Griewank function*, *Himmelblau function*, *noise function*.

**Example.** To make this concrete, let us suppose we want to compute the level sets of levels $\{-1, -0, 1, 2\}$ of the network $F\colon \mathbf{R}^D \to \mathbf{R}$. To compute the levels we add relu-layer to the network

$$R = \begin{bmatrix} 1 \\ 1 \\ 1 \\ 1 \end{bmatrix} \text{ and } r = -\begin{bmatrix} 2 \\ 1 \\ 0 \\ -1 \end{bmatrix} \tag{15}$$

derive the decomposition of the input space by the network, and collect all those 0-faces,...$(D-1)$-faces whose vertices are mapped to the corresponding level. Figure 11 shows four examples of level sets of networks.

### B.1.3 EXPERIMENTAL SETUP

**Introduction.** We took the Himmelblau function, the Griewank function, the quadratic function and the uniformly perturbed constant function as test function. We sampled 6400 data points and trained MLPs on this datar. Figure 12 depicts these fucntions.

**Himmelblau function.** The Himmelblau function has four local minima and one maximum. It has a step descent while outside of these extreme. We restricted it to the square $[-5, 5]^2$. It ranges between 0 and 890. Its local maximum is at $x = -0.3$ and $y = -0.9$, with 181.6 and its four minima are located at $(3, 2)$, $(-2.8, 3.1)$, $(-3.8, -3.3)$, and $(3.6, -1.8)$ with value 0.

$$f(x, y) = (x^2 + y - 11)^2 + (x + y^2 - 7)^2 \tag{16}$$

**Quadratic function.** We took the quadratic function restricted to the square $[-10, 10]^2$. We can use this function to study how well the networks pick up symmetry and curvature.

$$f(x, y) = x^2 + y^2 \tag{17}$$

**Griewank function.** We took the Griewank function restricted to the square $[-10, 10]^2$. Here we can study how the network captures the minima and maxima of the function.

$$f(x) = 1 + \frac{1}{4000} \sum_{i=1}^{2} x_i^2 - \prod_{i=1}^{2} \cos \frac{x_i}{\sqrt{i}} \tag{18}$$

**Noise function.** We uniformly sampled random points in the square $[-10, 10]^2$, the network has to learn the constant function.

### B.2 EFFECTS OF REGULARIZATION

### B.2.1 HIMMELBLAU FUNCTION

**Experimental details.** For the plots of Figure 14 we slowly increased $l_2$-regularization during training. All other hyper-parameters parameters (network: 2-50-50-1, relu-relu-linear activations, 6400 randomly sampled training data points from the [-5,5]x[-5,5] square, 1000 epochs, lr=1e-3, batchsize=128, ADAM, mse-loss, standard keras implementation) were fixed. After training we decomposed each network and derived the stars of each vertex. We only considered those decompositions which passed our tests: the volume should be equal to 100, the Euler characteristic should be 1, and no vertex should lie outside of the bounding input cube.

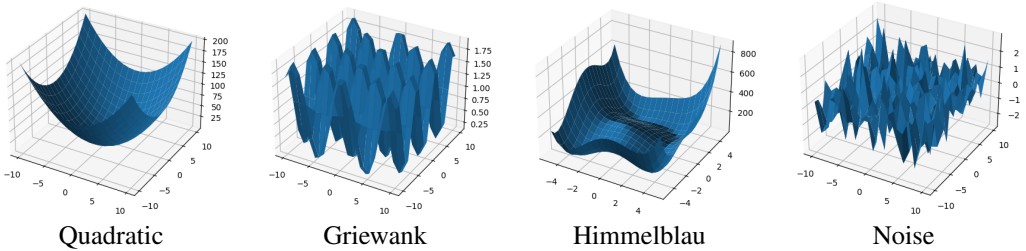

Figure 12: Plots of the functions used in the paper.

**Characteristics.** In these plots we show the computed characteristics: number of minima, number of maxima, span of the function, $l_2$-norm of the linear part of the function dubbed spectral bound, product of spectral norms of the weight matrices, maximal and minimal curvatures, final loss and final mse.

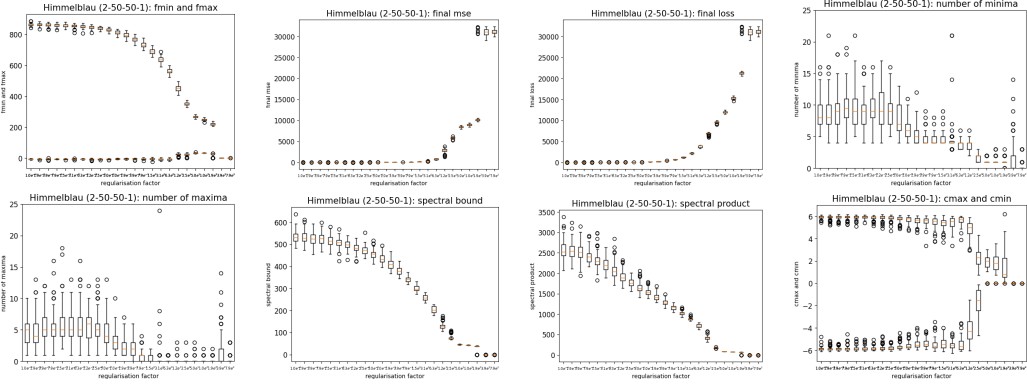

Figure 13: Derived characteristics of the Himmelblau while increasing regularization.

**Location of maxima and minima.** Figure 14 contains also extrema on the boundary. We colored minima red, maxima blue, and flat vertices in green.

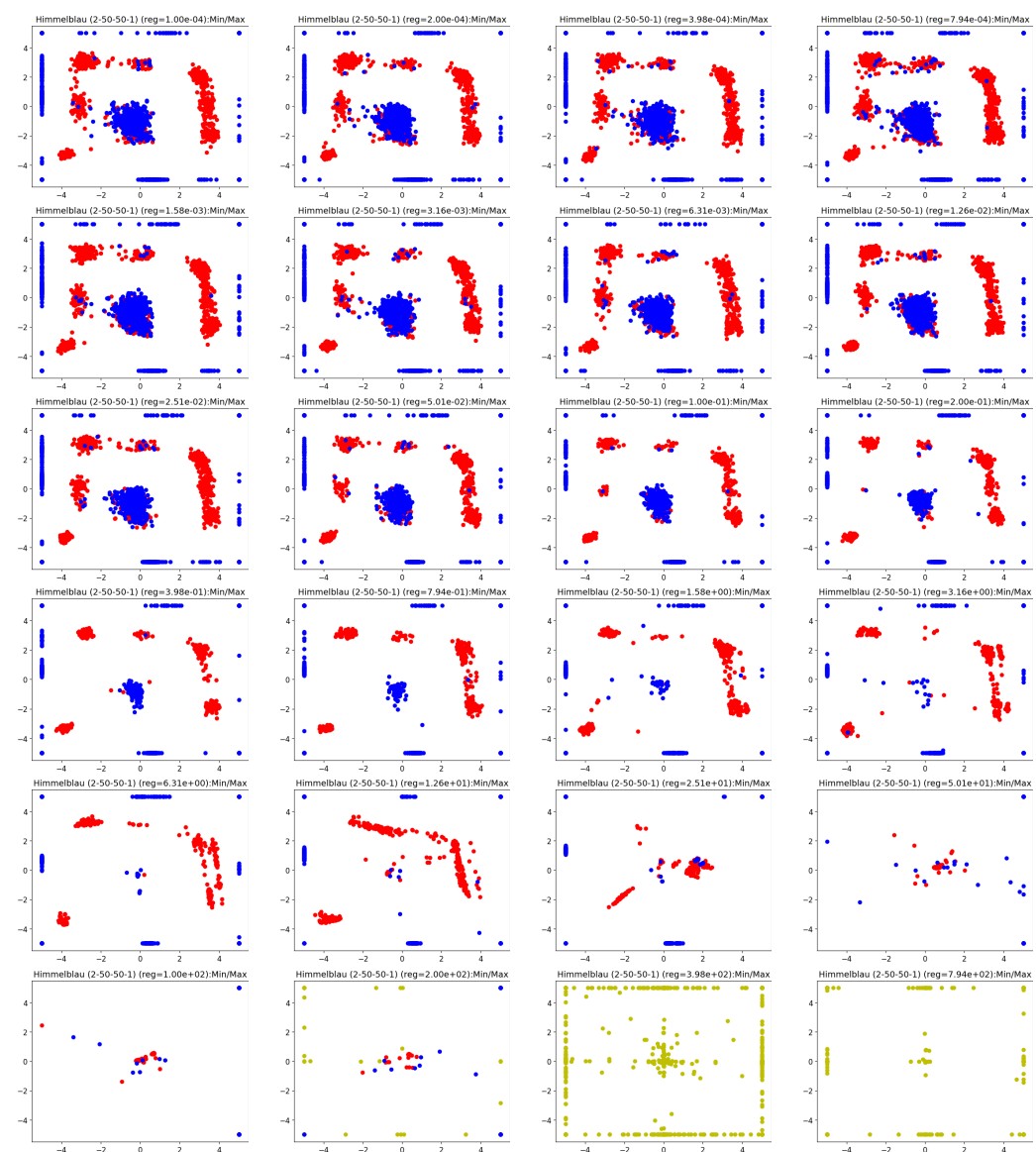

Figure 14: The locations of minima(red), maxima(blue), and flat points(green) while increasing regularisation. In each panel one-hundred experiments were run. Thus in total 2400 experiments.

**2d histograms.** For the histograms plots shown in Figure 16 and Figure **??** we only considered the inner extrema and excluded flat minima or maxima.

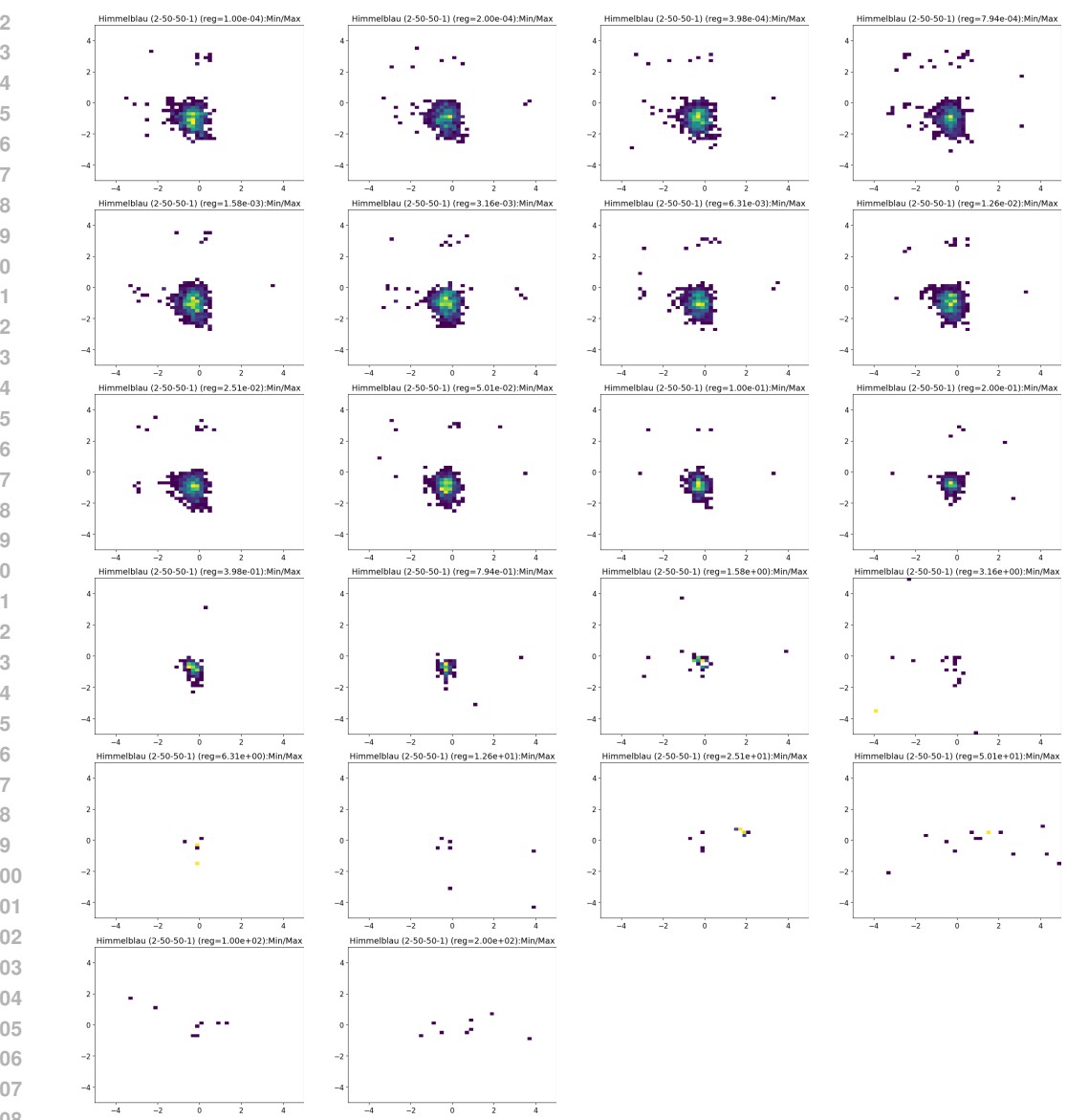

Figure 15: The locations and number of maxima of the Himmelblau function while increasing regularization. In each panel one-hundred experiments were run.

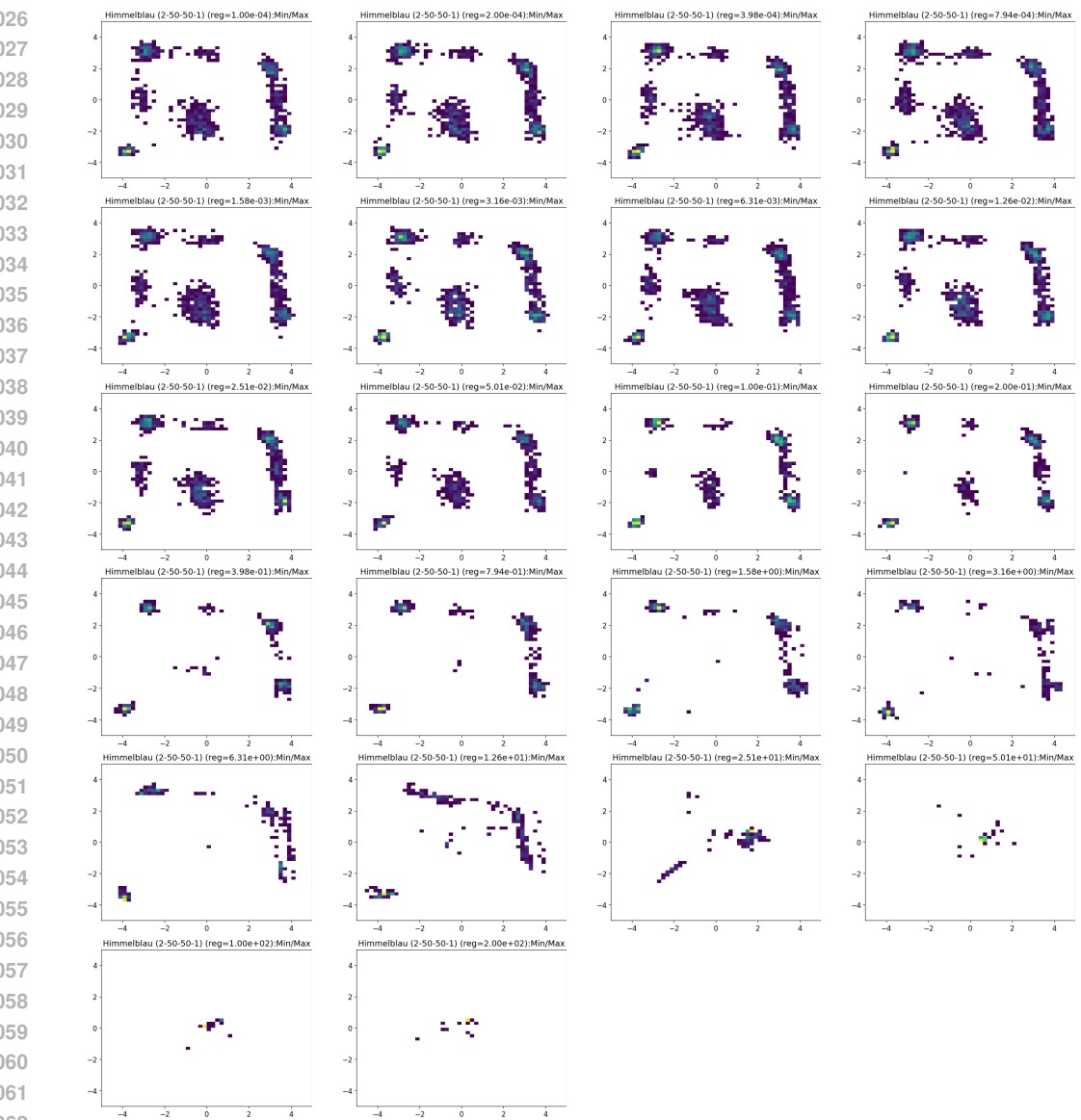

Figure 16: The locations and number of minima of the Himmelblau function while increasing regularization. In each panel one-hundred experiments were run.

**Discussion.** We were surprised how big we had to increase the regularization factor until the training process broke down. This explains the small increases of the regularisation factor.

We can see that the networks capture the essence of the Himmelblau function, although the location (and number) of minima varies a lot. Only after we increased regularization a lot the number location of the extrema become more correct. But the price we pay is that the mse of the network becomes faulty. In summary one has to balance what is the intention of the function. Either good mse but bad localization or good localization and worse mse.

### B.2.2 GRIEWANK FUNCTION

**Experimental details.** For the plots of Figure 17 we slowly increased $l_2$ regularization during training. All other hyper-parameters parameters (network: 2-20-20-1, relu-relu-linear activations, 6400 randomly sampled training data points from the [-10,10]x[-10,10] square, 1000 epochs, lr=1e-3, batchsize=128, ADAM, mse-loss, standard keras implementation) were fixed. After training

we decomposed each network and derived the stars of each vertex. We only considered those decompositions which passed our tests: the volume should be equal to 100, the Euler characteristic should be 1, and no vertex should lie outside of the bounding input cube.

**Evaluation.** The first set of plot contains also extrema on the boundary. We colored minima red, maxima blue, and flat vertices in green.

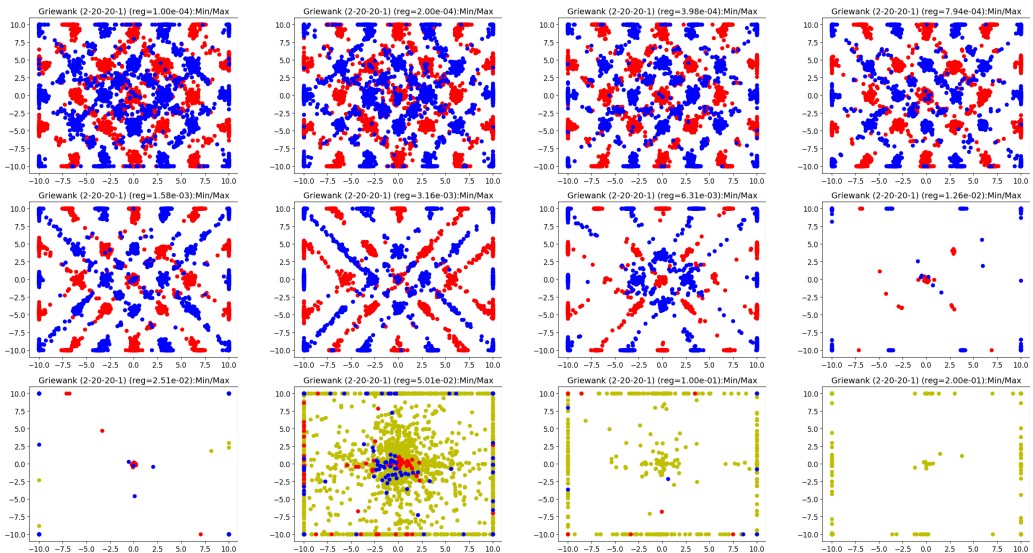

Figure 17: The locations of minima(red), maxima(blue), and flat points(green) while increasing regularisation. In each panel one-hundred experiments were run. Thus in total 1600 experiments.

**2d histograms.** For the histogram plots we only considered the inner extrema. The sixteen panels shown in Figure 18 encode the number of minima and locations of the Griewank function as regularization is increased. The sixteen panels shown in Figure 19 encode number and locations of maxima of the Griewank function as we increase regularization.

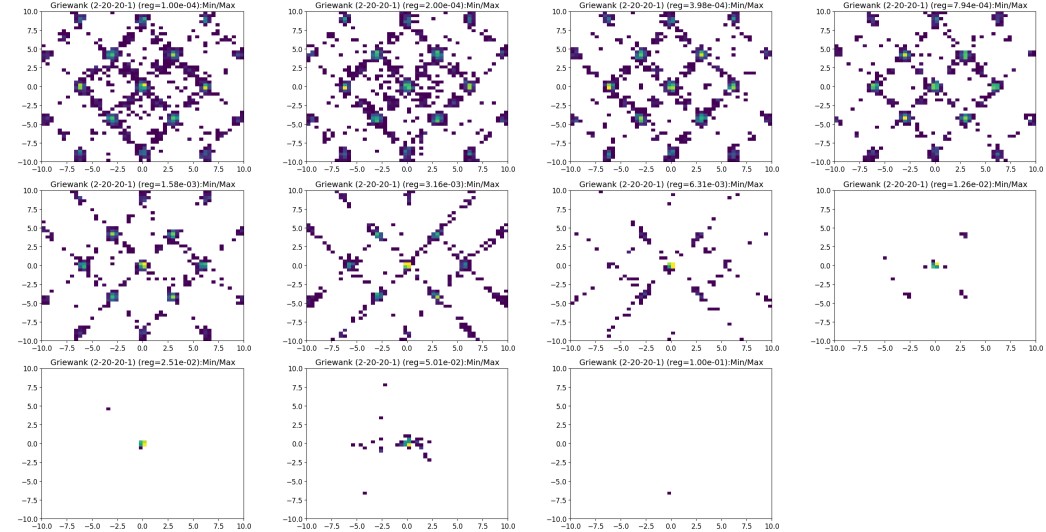

Figure 18: The location in number of minima in a 2d-histogram plots while increasing regularization. In each panel one-hundred experiments were run, but we only considered those experiments for which algorithm outputs a valid decomposition. There are no local minima for a regularization factor of $7.94e - 01$.

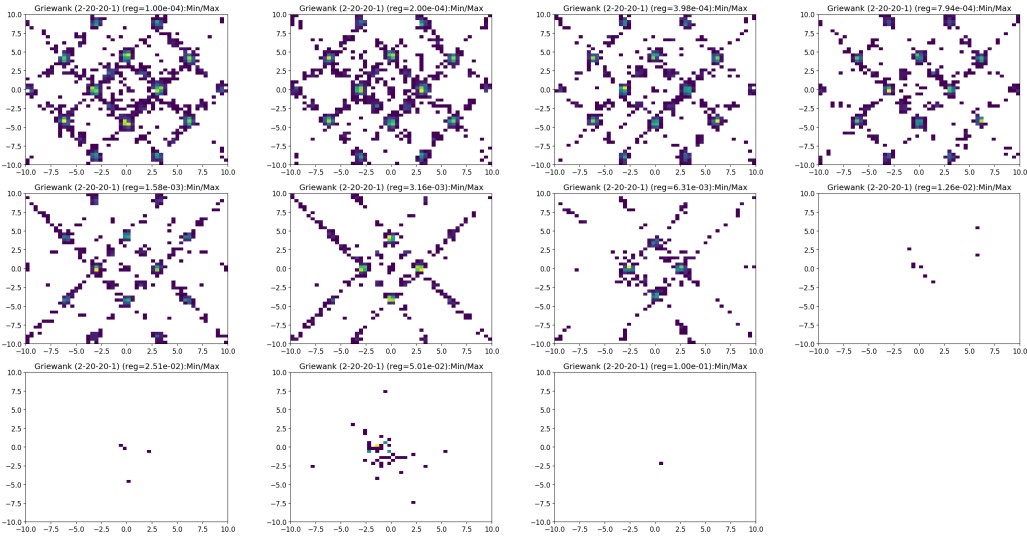

Figure 19: The location in number of maxima in a 2d-histogram plots while increasing regularization. In each panel one-hundred experiments were run, but we only considered those experiments for which algorithm outputs a valid decomposition. There are no local maxima for a regularization factor of $7.94e - 01$.

## B.3 EFFECTS OF DEPTH AND WIDTH

### B.3.1 INCREASING DEPTH

**Himmelblau functions.** For the plots of Figure 22 we started with a network of type 2-20-1 and increased the depth up to networks of type 2-20-20-20-20-20-20-1. All other training parameter (no regularisaiton, relu-...-relu-linear activations, 6400 randomly sampled training data points from the [-5,5]x[-5,5] square, 1000 epochs, lr=1e-3, batchsize=128, ADAM, mse-loss, standard keras

implementation) were fixed. After training we constructed the level set networks of four of these networks and computed the decomposition as well.

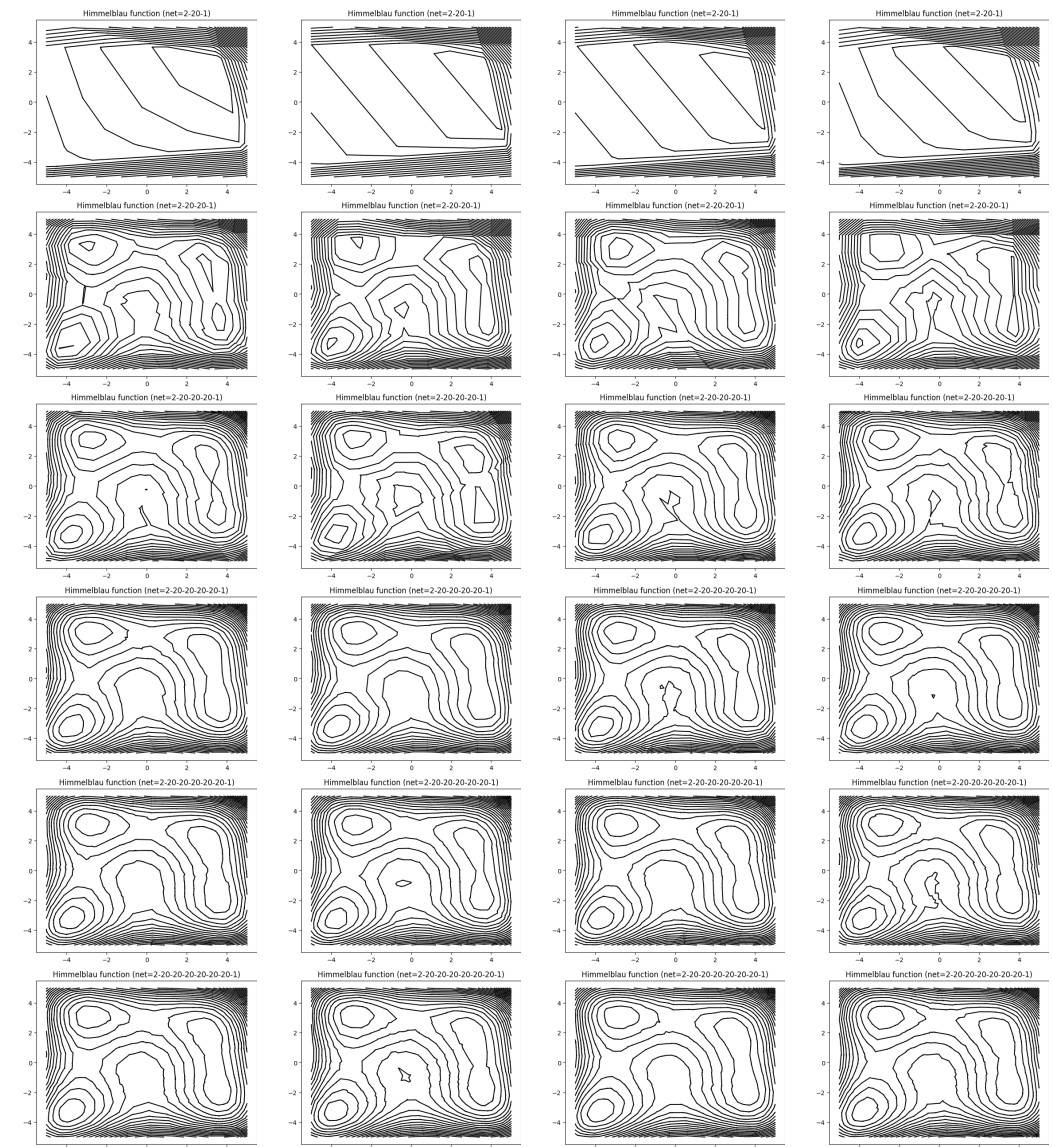

Figure 20: Level sets of networks trained on the Himmelblau function with increasing depth. Each row has the same depth.

**Discussion.** Again we see that in our setting we need at least two layers to capture the Himmelblau function. Futher it is obvious that this is a difficult function for the network to capture. Visually, it looks convenient for networks of depth six, but the network struggles to capture the curvature correctly.

### B.3.2 INCREASING WIDTH

**Himmelblau functions.** For the plots of Figure 23 we started with a network of type 2-5-5-1 and ended with networks of type 2-50-50-1. All other training parameter (no regularization, relu-...-relu-linear activations, 6400 randomly sampled training data points from the [-5,5]x[-5,5] square, 1000 epochs, lr=1e-3, batchsize=128, ADAM, mse-loss, standard keras implementation) were fixed.

After training we constructed the level set networks of two of these networks and computed the decomposition as well.

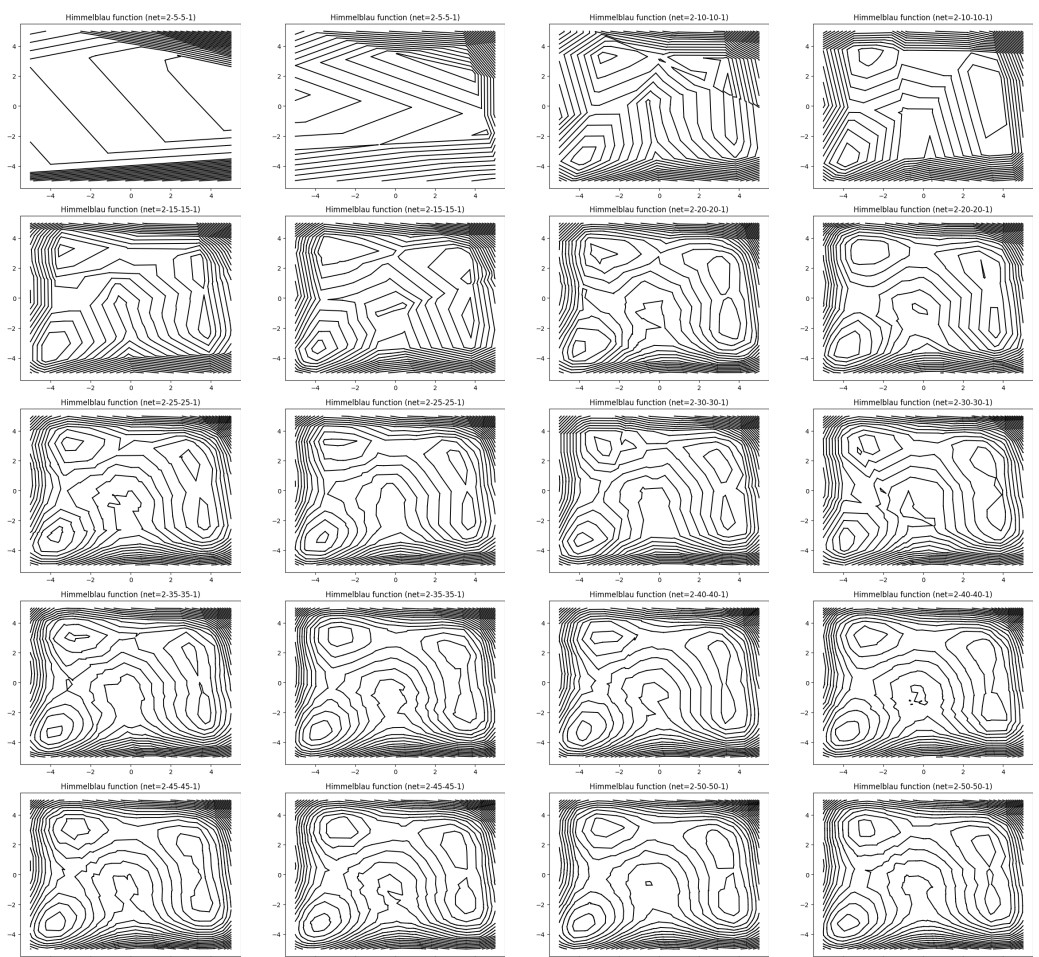

Figure 21: Level sets of networks trained on the Himmelblau function with increasing width.

**Discussion.** We see that even with 50 neurons and two hidden layers the curvature is not captured.

### B.3.3 INCREASING DEPTH

**Griewank functions.** For the plots of Figure 22 we started with a network of type 2-20-1 and increased the depth up to networks of type 2-20-20-20-20-20-20-1. All other training parameter (no regularisaiton, relu-...-relu-linear activations, 6400 randomly sampled training data points from the [-10,10]x[-10,10] square, 1000 epochs, lr=1e-3, batchsize=128, ADAM, mse-loss, standard keras implementation) were fixed. After training we constructed the level set networks of four of these networks and computed the decomposition as well.

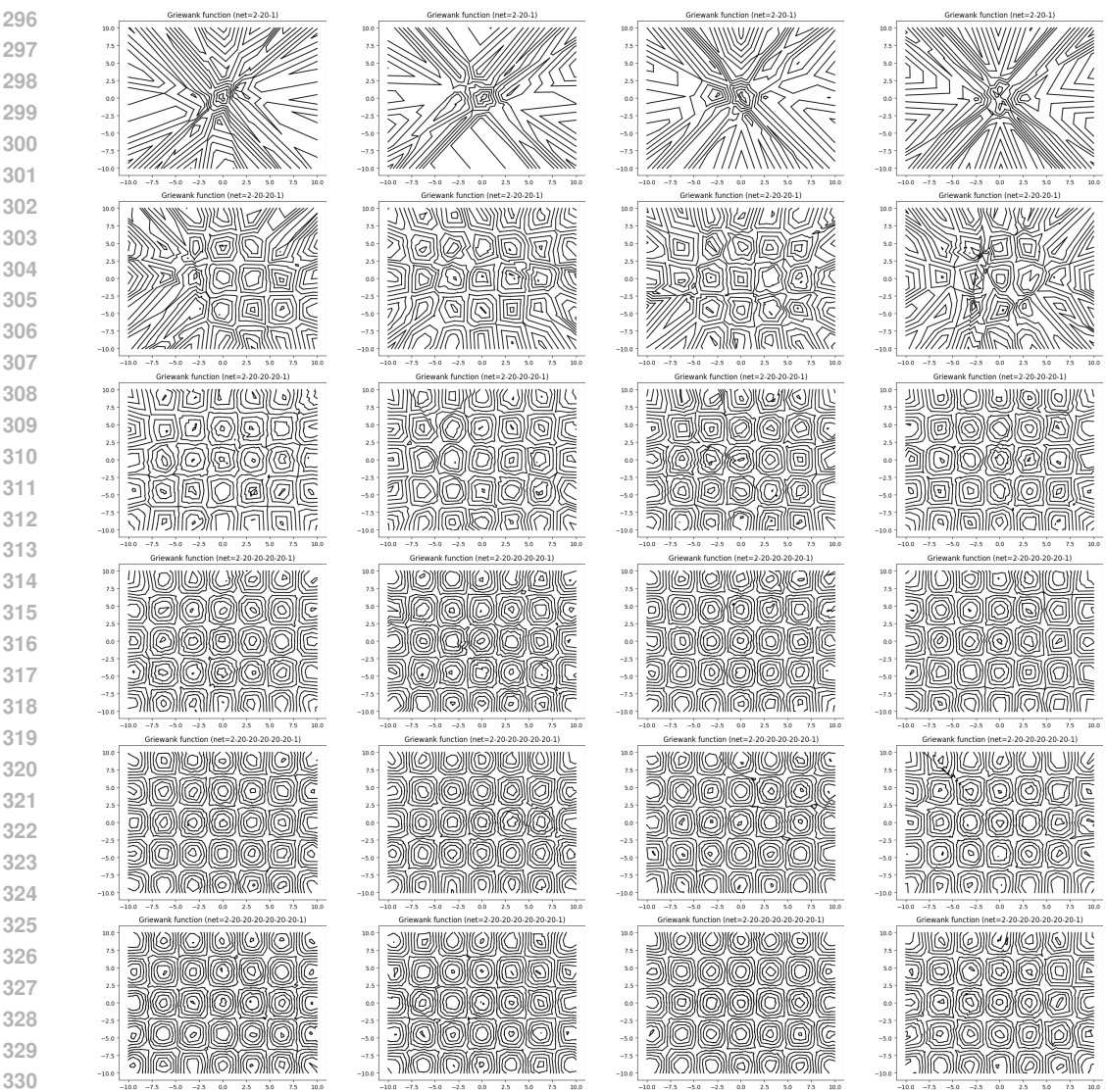

Figure 22: Level sets of networks trained on the Griewank function with increasing depth. Each row has the same depth.

### B.3.4 INCREASING WIDTH

**Griewank functions.** For the plots of Figure 23 we started with a network of type 2-5-5-1 and ended with networks of type 2-50-50-1. All other training parameter (no regularization, relu-...-relu-linear activations, 6400 randomly sampled training data points from the [-10,10]x[-10,10] square, 1000 epochs, lr=1e-3, batchsize=128, ADAM, mse-loss, standard keras implementation) were fixed. After training we constructed the level set networks of two of these networks and computed the decomposition as well.

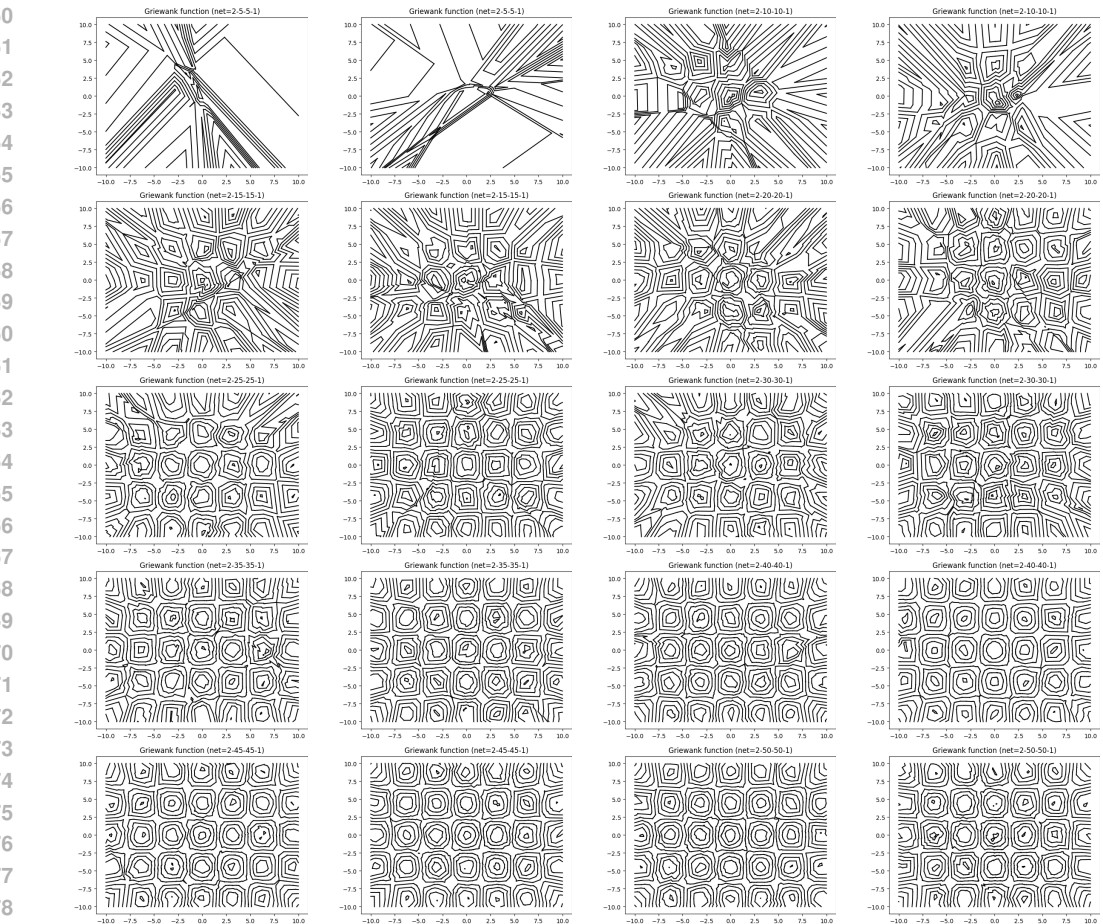

Figure 23: Level sets of networks trained on the Griewank function with increasing width.

**Discussion.** We see that even with 50 neurons and two hidden layers the curvature is not captured.

# C    FURTHER DETAILS ON THE ALGORITHM

## C.1    ALTERNATIVE ALGORITHMS

**Related algorithms.** For small networks Balestriero & LeCun (2023) provides an algorithm which returns the exact number of cells and their activation pattern. It has some similarities to our algorithm but differs in scope. More crucially, it keeps track of the current decomposition by storing the activation patterns of non-void cells. In a sense this algorithm operates entirely on the h-representation of the polytopes. In contrast, we keep track of the cells by storing its vertices. In fact we use both v- and h-representation of the polytopes. The vertices are necessary to derive the stars, which in turn capture the local behavior of the network.

Berzins (2023) provides an algorithm for the enumeration problem of neural networks. Their core idea is a sequential cutting of the 1-skeleton of the complex per layer. So at a layer, all potential hyperplanes are generated by going through all possible activations patterns of the neurons. If the hyperplane cuts an edge, the sign pattern of the vertices of the edge differ and we get two edges and a novel vertex. In addition one has to consider 2-faces as the intersection with the hyperplane gives a novel edges as well. There is some similarity to step $(\alpha^3)$ of our algorithm but our algorithm derives the entire structure of the polytopal complex not just the 1-skeleton. This cell structure is necessary to derive curvature and volume.

Another work presented in Humayun et al. (2024a), Humayun et al. (2023) provides an impressive algorithm to compute the polyhedral complex of a two dimensional network and its decision boundaries. But the algorithm presented in Humayun et al. (2023) works only for two dimensional inputs - furthermore it seems challenging to extend the core idea of the algorithm to higher dimensions. Our algorithm does not have a dimensional restriction. We further complement the subsequent work on grokking of Humayun et al. (2024b) by analysing the effect of regularisation on the distribution of cells.

## C.2 Pitfalls

**Numerical issues.** Partitioning input spaces by neural networks may lead to numerical issues. The main culprit are imprecisions caused by floating numbers. We identified three different causes. Our algorithm partitions the polytopes of layer $k$ separately. This implies that we compute vertex as the intersection of a set hyperplanes several times. Each time these hyperplanes are not the same. While attaching polytopes in step $(\beta)$ of the algorithm we have to decide if two of such vertices are in fact the same. Another error can be introduced while computing the affine dimension of a set of vertices - we do this based on the svd. Finally while cutting a polytope by a hyperplane, we determine if the vertices of the polytope lie in that hyperplane. Further we intersect the hyperplane with the links of the polytope. Again we have to decide if the point of intersection lies in the polytope. To do this we compute the sign of the vertex with respect to the hyperplane - this can be numerically challenging if we are close to zero.

Most numerical issues arise for almost similar weights. This causes almost identical features which give rise to arbitrary close vertices while intersecting another feature.

We observe this during heavy regularization. Here the network tend to cluster the weights leading to sets of almost equal weights. But also while constructing examples one tends to overdetermined the vertices leading to challenging problems.

For our implementation we increased the number of precision to 30 while doing internal computations using the `mpmath` library. Furthermore, computing the Euler characteristics and volume considerations give further evidence of the validity of a decomposition. Let us remark that also mention numerical difficulties and resolve this by going to double precision on their `GPU` implementation.

**Computational resources total number of experiments.** We trained and decomposed many networks during the preparation of this paper. Although the networks are small their shear number makes it hard to repeat this in reasonable time. A small network (2-5-5-1) is decomposed in a few seconds at most, the larger ones (2-100-100-1) can take up some time maybe 30minutes. Particularly time consuming is the computation of the levels sets. We did all of this on an internal cluster on an CPU, implemented in pure python. We estimate the number of experiments of the main paper to be 5000. The additional experiments in the appendix about another 10000. A single experiment can be done easily on a personal pc.

