# OpenReview forum: "The polytopal complex as a framework to analyze multilayer relu networks"
_ICLR.cc/2025/Conference — Submitted to ICLR 2025_

### Official Review · Reviewer_Att1 · 2024-10-22

**Soundness:** 3
**Presentation:** 3
**Contribution:** 2
**Rating:** 5
**Confidence:** 3

**Summary:**

This paper considers neural networks with linear layers and ReLU activation function. In this case, the network is a continuous piecewise linear function and the input space can be decomposed into cells in which the network is an affine function for each cell. The authors provided an algorithm to compute the polytopal complex formed by such cells of a neural network. By this decomposition, they can compute several statistics, such as the maxima, minima, number of cells, local span, and curvature of the network. They also provide several empirical results for some functions such as the Himmelblau function and the Griewank function.

**Strengths:**

1. This paper provides a novel algorithm to compute the polytopal complex of a neural network.
2. From the obtained polytopal complex the authors can analyze several properties, such as the maxima, minima, number of cells, local span, and curvature of the network.
3. The authors also analyze the effects of depth, width, and regularization on the complex.

**Weaknesses:**

1. The time complexity $O(|vertices|)$ of the algorithm in this paper is very high, since the number of vertices obtained in this algorithm should be some exponential functions of the number of neurons or the number of layers in the network, which makes the algorithm not very useful in practice, especially for deep networks.

2. The assumption that the network is a continuous piecewise linear function is also not very useful since the DNNs used in practice have much more complicated structures.

**Questions:**

1. It will be useful if the authors can provide more analysis on the bounds of the number of vertices obtained in their algorithm, thus providing more information on the complexity of their algorithm.
2. In Equation (1), it should mention that $\sum \lambda_i = 1$, otherwise it is not true.
3. More references on the bounds of the number of cells and vertices should be added to the paper.

**Details Of Ethics Concerns:**

No.

---

> ### Author Response · Authors · 2024-11-27
>
> Dear Reviewer,
>
> Thank for taking the time to read our paper and your suggestions. We revised the paper, notably moving Section 4.1 to the appendix and added a new Section 4.1 which performs some generalization experiments. Moreover, we added some references in the Literature section.
>
> *The assumption that the network is a continuous piecewise linear function is also not very useful since the DNNs used in practice have much more complicated structures.*
>
> Most ReLu feedforward networks lead continuous piecewise linear function. We do not know of any statistics about the fraction of ReLU networks used in practise, but we firmly believe that they are being used in real world settings. You are right that not all networks, eg. transformers, lead to pl-functions.
>
> *On complexity*
>
> Add your suggestion we added more known bounds and references on the number of linear regions. Unfortunately, we did not manage to do this before the upload-freeze. You are right that the number of regions grows exponentially, as the input dimension increases. Empirically, the most crucial parameter for complexity was the number of neurons in the first layer. Depth did matter – but we managed to decomposed networks of depth 30. Also, the literature provides decomposition examples for deep networks. We can cast the vertex enumeration problem as a MLP, basically setting the hyperplanes as the neurons of the first layer. The best-known algorithm has complexity of O(vnd) so it would be rather surprising if we could beat it. We would argue that we only compute what is necessary – we will add a graphical explanation of the steps of the algorithm together with an upper bound of the vertices we obtain in this way.

---

### Official Review · Reviewer_nVeW · 2024-10-28

**Soundness:** 2
**Presentation:** 2
**Contribution:** 2
**Rating:** 5
**Confidence:** 4

**Summary:**

This paper is motivated by the fact that not all activation regions contain training data. The work can be divided into two parts. In the first part, the authors introduce a variation of Region Subdivision algorithm that allows them to use both H- and V-representations of the cells in the polyhedral complex produced by ReLU networks. In this part they also provide a thorough analysis of the algorithm, most notably, in regards to validity, and timing. In the second part, they leverage the obtained decomposition for various analyses, such as analyzing the cell volume, star volume and curvature. They also analyze the impact of width, depth, and regularization parameters on the computation time and number of cells.

**Strengths:**

1. The paper reads well. The flow between the sections and paragraphs is smooth. Some Figures need minor improvement for better clarity, but in overall they are well thought through. I particularly like the usage of level sets, as they make visualizing the three dimensional functions much clearer, and, as far as I know, it's not a very common approach in this field.
2. I really enjoyed Section 3.2. It is absolutely necessary to validate the polyhedral complex obtained by our algorithms, yet, as far as I know, this is the first work that actually mentions the steps taken to ensure validity. This is a good step towards more trustworthy methodologies.
3. Great analysis of the decomposition time in Figures 5 and 6. It is known that computing the polyhedral complex is tremendously computationally intensive, yet not many works provide detailed runtime analysis-only other work I know of that does something similar is the work of Serra et al. (2018), although their analysis is less detailed.
4. As far as I know, this is the first work to show the impact of regularization on the number of activation regions.
5. The motivation behind the paper is really interesting. I agree with the authors that there is a “need for methods which extend the validity of networks beyond the test data”. This also fits the data-driven principles perfectly, and might allow for more informed data augmentation/pruning strategies in the future if this method is extended to real world applications.

**Weaknesses:**

# Inconsistent story

In the *Motivation* paragraph of Section 1 the authors mention that this paper is motivated by a need for methods that extend the validity of networks beyond the test data. Despite that, this is not the main focus of the later sections. It appears to me that the motivation does not match the paper. Authors later state that “this paper extends the validity of a neural network beyond a discrete test data point to its neighbors”. I don’t believe that this paper actually meets that claim. After reading the *Introduction* I expected to see experiments showing me how we can perform testing beyond the test set, and how it changes the perceived generalization capabilities of a model.  However, there are no such experiments in this work. To me, the paper focuses more on investigating properties of linear regions, rather than extending testing beyond the test set.

I expect the authors to rewrite the Introduction so that it fits the rest of the paper, and doesn’t make any false claims. To reiterate, in its current form, the paper only shows that it is theoretically possible to extend the testing beyond the test set, and that stars of a polyhedral complex could be used for that. However, there is no explicit algorithm proposing this extension, neither are there any experiments showcasing the validity of that extension, despite Section 1 hinting that it's the main focus of the paper,

# Poor literature review

The literature review of the field of linear/activation regions is practically nonexistent. The authors missed several essential works from the field of activation/linear regions. Below I list the most influential ones that I would expect to be referenced by any paper in this field.

[1] Hanin, B., & Rolnick, D. (2019, May). Complexity of linear regions in deep networks. In International Conference on Machine Learning (pp. 2596-2604). PMLR.

[2] Wang, Y. (2022, July). Estimation and Comparison of Linear Regions for ReLU Networks. In IJCAI (pp. 3544-3550).

[3] Liu, Y., Cole, C. M., Peterson, C., & Kirby, M. (2023, September). ReLU neural networks, polyhedral decompositions, and persistent homology. In Topological, Algebraic and Geometric Learning Workshops 2023 (pp. 455-468). PMLR.

[4] Arora, R., Basu, A., Mianjy, P., & Mukherjee, A. (2018). Understanding deep neural networks with rectified linear units. ICLR.

[5] Serra, T., Tjandraatmadja, C., & Ramalingam, S. (2018, July). Bounding and counting linear regions of deep neural networks. In International conference on machine learning (pp. 4558-4566). PMLR.

[6] Raghu, M., Poole, B., Kleinberg, J., Ganguli, S., & Sohl-Dickstein, J. (2017, July). On the expressive power of deep neural networks. In international conference on machine learning (pp. 2847-2854). PML.

[7] Novak, R., Bahri, Y., Abolafia, D. A., Pennington, J., & Sohl-Dickstein, J. (2018). Sensitivity and generalization in neural networks: an empirical study. ICLR.

[8] Gamba, M., Chmielewski-Anders, A., Sullivan, J., Azizpour, H., & Bjorkman, M. (2022, May). Are all linear regions created equal?. In International Conference on Artificial Intelligence and Statistics (pp. 6573-6590). PMLR.

[9] Hanin, B., & Rolnick, D. (2019). Deep relu networks have surprisingly few activation patterns. Advances in neural information processing systems, 32

[10] Zhang, X., & Wu, D. (2020). Empirical studies on the properties of linear regions in deep neural networks. ICLR.

[11] Croce, F., Andriushchenko, M., & Hein, M. (2019, April). Provable robustness of relu networks via maximization of linear regions. In the 22nd International Conference on Artificial Intelligence and Statistics (pp. 2057-2066). PMLR.

[12] Xiong, H., Huang, L., Yu, M., Liu, L., Zhu, F., & Shao, L. (2020, November). On the number of linear regions of convolutional neural networks. In International Conference on Machine Learning (pp. 10514-10523). PMLR.

## Minor issues stemming from poor literature review

1. [3] already showed that adjacent activation regions differ in only one bit of their activation sequence, which the authors mention in L232, and so should be correctly cited.

2. Until Section 3 it is unclear if the authors work on activation or linear regions. $C_k = \text{conv} (v_1, ..., v_p\)$ requires convexity, which doesn’t hold for linear regions as mentioned by [9], so for clarities sake authors should clarify which regions they focus on early in their work.

# Novelty
Frankly, I am unsure whether the paper is novel enough to be accepted to a venue like ICLR. The algorithm in Section 3.1 is an incremental modification of the classical Region Subdivision algorithm. The validity and time checks are a pleasant addition to the paper (compared to relevant literature), but they do not provide significant novelty. Similarly, other contributions that I praise in *Strengths* are new but do not feel novel enough to me to deem acceptance to ICLR.  My opinion on novelty would significantly change if the authors performed experiments in which they “extend the discrete training data set to a neighborhood given by the union of the cells, and show experimental results”, rather than showing that it is theoretically possible. I believe that this would be a very strong and novel contribution. Especially, if the authors managed to generalize this outside of toy datasets (possibly by employing approximations or taking a scenario from embedded systems or virtual sensors mentioned in Section 6). I believe that without this, the work would not be of interest to the wider research community.

Consequently, a possible future direction that the authors can take is proposing and implementing an algorithm that extends testing beyond the test set. Analytically computing the polyhedral complex in large datasets (or even on MNIST) is absolutely unfeasible. However, I think that the authors could estimate the neighboring linear regions using linear search (monitoring changes in activations along a random vector from a data point). This would allow them to extend both the training and test sets. Both could be used for measuring robustness, while the latter could be used for achieving more thorough accuracy (it’s important to find if incorporating these new test point has any effect on the perceived accuracy and robustness though).

# Potential improvements in clarity (minor)
Here I propose few changes  that could be implemented to further improve clarity (please consider these as simply suggestions rather than requests for change):
 1. Not all readers will be acquainted with topology, and visualizing what a star is would allow for easier reading.
 2. There are typos in Lines 134, 150, 235-236, 265, 266
 3. In L150 authors do not mention which appendix to go to.
 4. Wouldn’t it be easier to visualize the Himmelblau and Griewank functions rather than explaining them?
 5. The Figures 4 and 5 don’t specify the unit for time. Figure 4b has the wrong ylabel. The digits in the yellow cells of Figure 6 are unreadable.
 6. What are $\rho$ and $\nu* from L128?
 7. It would be great to provide a few sketches that would simplify understanding of the algorithm from Sec. 3 for readers that are new to the field, especially for ($\alpha^3$) which is very confusingly written.

# Summary
I believe that in its current state the paper should be rejected. However, if the authors address my issues regarding *Inconsistent story* and *Poor literature review* I am happy to increase the rating of the paper. However, my rating is unlikely to change beyond boundary reject (5). In my opinion, for this paper to introduce a strong contribution to the research community the authors should expand it towards extending the testing beyond the test set with some experiments showing the applicability of this new technique (ideally beyond toy datasets).

**Questions:**

# Clarification
1. What did the authors mean in L369-370 (“Further, looking at … interpolating the data.”)?
2. What did the authors mean by “until the training collapses” in L414?
# Curiosity
1. The idea behind motivation made me think about robustness. I know that currently the setting is closer to regression than classification. Have the authors thought about generalizing to classification? I think it would be interesting to see the relation between the robustness of an activation region and the number of samples it contains (as I mentioned in the Weaknesses when proposing a possible future direction).

---

> ### Author Response · Authors · 2024-11-25
>
> Dear reviewer,
>
> thank you reading our paper, the questions and suggestions. We revised the paper. The new version has an updated literature review and a novel Section 4.1. We moved the old Section 4.1 to the appendix. In Section 4.1(new) we provide two experiments addressing generalisation. Thanks again for making us think about the paper and its story.
>
> As a small rebuttal, we don't think that the two-dimensional functions are toy examples, they are quite challenging to train for the network. The Griewank and the Himmelblau functions are designed to test optimization algorithms. Suppose, we use an NN as a surrogate model to find minima, looking at the appendix, not all architecture excel at this.
>
> **Clarification**
>
> *What did the authors mean in L369-370 (“Further, looking at … interpolating the data.”)?*
>
> At your suggestion we moved the part to the appendix. To explain what we mean, and how it relates to curvature, let us think in the 1d-setting. In 1d we are looking at a continuous piecewise affine function defined over intervals. Now in some of these intervals data points are given. Assuming a small training error, the network interpolates these points sufficiently enough. In between the good intervals may be some from which we do not know anything. Now, let us look at curvature: If the curvature is small, this means in the pl-setting, that the difference between the linear functions defined on two neighbouring segments is small (as measured on S^1 for example). If we go from an interval with data to the next intervals with data and pass only intervals with small curvature changes, the network is basically almost linear and we interpolate the data. In contrast, if the curvature is large, basically anything can happen between these two intervals.
>
> * What did the authors mean by “until the training collapses” in L414?*
>
> Here we mean that the network did not converge, the hyperplanes started to cluster, becoming very similar thus causing numerical troubles for our algorithm.
>
> **Curiosity**
> *The idea behind motivation made me think about robustness. I know that currently the setting is closer to regression than classification. Have the authors thought about generalizing to classification? I think it would be interesting to see the relation between the robustness of an activation region and the number of samples it contains (as I mentioned in the Weaknesses when proposing a possible future direction).*
>
> We have thought about classification more coming from a topological viewpoint. What you suggest is interesting, let us write down what we observed.
>
> -	There is no general rule how the network classifies the input, most often it does so by a linear classifier in a cell. Appendix B.1.1 shows how to extract these cells. In the spirit of your curiosity one can check this classifier. The same applies to your suggested exploration algorithm. But, we also observed “bended” hyperplanes as classification bounds. Our preliminary explanation for this is that the initialisation defines some hyperplanes which separate the data along classes, but during SGD-training only the linear classifiers are optimized, the positions of the hyperplanes do not contribute to the gradient. If we want to analyse robustness, we must measure the distance to both.
>
> Some challenges lie ahead:
>
> -	the numerics of classification networks are way more challenging than for regression networks. Doing the classification trick of Appendix B.1.1 adds a type of Braid-arrangement to the network. Here many vertices are overdetermined, causing floating point troubles.
>
> -	As already noted by you, the number of linear regions explode, we observed many tiny cells in typical MNIST classification networks, they do not contribute to the classification (their volume is basically zero) our algorithm must keep track of them. An explorative algorithm will face similar problems: It is hard to find a interior point of a small polytope, qhull will throw an error (If we want to get the vertices form the h-representation). In dimension bigger than 10 the number of vertices of a single cell will become unmanageable.
> In summary we have peak into these structure by looking at paths in the input space, or hyperplanes, or bigger substructures. In order to do so we need a solid basis, we hope the paper provides such.

---

> > ### Comment · Reviewer_nVeW · 2024-11-25
> >
> > Dear Authors,
> >
> > Thank you for your response, and for addressing my concerns. I noticed the updated literature, and although no change in tone in Introduction regarding *Motivation* was made, the new Sec. 4 answers some (although not all) of my worries. I feel like the quality of the paper has improved and so I have raised my mark to 5 as promised. After reading your responses and the revised paper I have the following comments.
> >
> > **Duality of the motivation**
> >
> > In the paper you use classification example for motivation. From this perspective, using 2D data is too simple of a problem, and, understandably, makes the reader curious about applications to more real-world-like data (MNIST at minimum). However, based on the responses you provided to me and Reviewer BhtF, I understand that you are more interested in surrogate modelling. These two are completely different problems, and so, for the sake of clarity, no mention of classification should be made in the paper (unless to explicitly specify that this work does not concern itself with classification). Furthermore, the field of surrogate modelling should be introduced in this paper more concretely, and relevant literature should be included to inform the reader about the novelty that this work brings to this area.
> >
> > To elaborate, your work falls into three boxes: 1) introducing new method; 2) measuring new properties of linear regions; and 3) connecting the field of linear regions with other field(s). In my opinion, (1) is an incremental improvement, (2), in its current state, is not enough for an ICLR submission, and (3) was barely mentioned (although it's in significantly better state than the initial submission). (1) cannot be improved upon. Similarly, I don't think that (2) can be significantly improved upon, as it's either incremental (Sec. 3.2, 3.3 & 3.4) or its main purpose is to advertise (3) (App. B1). Hence, (3) feels like the most promising categorisation to me. However, the paper cannot be said to be mainly about (3) - there is no introduction of other field(s), and no explanation about the gaps in these field(s) that could be filled with linear regions.
> >
> > Unless my understanding of your motivation behind this paper is wrong, I believe that (3) is the direction you should go with in regards to this work.
> >
> > **Clarity**
> >
> > - What is *dev* in L323? It needs to be defined properly. Similarly, the difference between the *train* and *true* needs to be made clear.
> >
> > **Summary**
> >
> > I am happy that the authors have listened to my feedback, and I believe that the quality of the paper has improved, although I still do not believe that it's good enough to be accepted to ICLR. The story told by the authors is not consistent, with the motivation not being fully addressed by the authors.

---

> > > ### Author Response · Authors · 2024-11-27
> > >
> > > Dear reviewer,
> > >
> > > thank you for your feedback and your box-model, we think we got your point. The intention of the classification plot was to wet the appetite for the need to look at the decomposition, hence the algorithm. But we see your point that it mad you hungry for results on classification which we provide none. For now we will make it clear in the introduction/motivation.
> > >
> > > We have another question: We believe that testing a data point in a cell, basically checks the cell (of course oversimplified) and it might check the neighbouring cells as well. If there is some truth to this, then we could argue that we go from the discrete to the continuous, covering parts of the volume as being correct. What are we missing in this argument?

---

> > > > ### Comment · Reviewer_nVeW · 2024-11-28
> > > >
> > > > Dear Authors,
> > > >
> > > > That's a good question, and I guess it depends on the setting. In classification, which I am more familiar with, the decision boundary is defined as the intersection between the max output functions. Consequently, the decision boundary divides some of the linear regions via the intersections of max functions. Hence, there will be majority of linear regions that contain no decision boundary, but there will be also a minority containing the decision boundary. For the latter, unlike for the former, testing a data point in a cell does **not** check the cell. So, you could make that argument for linear regions far from the decision boundary, at which point the finding is not very useful. However, making the same argument for linear regions close to decision boundary is problematic, and is already something that is being done (although without the use of linear regions) in the field of adversarial examples (*Certifiably Robust Training*). So you would have to compare the cost/benefit ratio of your approach to the ones used in that field.
> > > >
> > > > When it comes to regression I am not sure. Frankly, I don't know what testing a data point in a cell entails in this setting, and I am not sure how useful it might be as my experience in this setting is very limited.
> > > >
> > > > I hope this answers your questions. If not, then we still have some time for further discussion.

---

### Official Review · Reviewer_aFh6 · 2024-10-29

**Soundness:** 2
**Presentation:** 3
**Contribution:** 2
**Rating:** 3
**Confidence:** 2

**Summary:**

This paper proposes an algorithm that decomposes the input space of a ReLU MLP in convex polytopes. This algorithm allows for analyzing such neural networks beyond validation points, including properties such as curvature, hyperplanes, and stars.

**Strengths:**

Interesting research direction; Figure 8 looks awesome.

**Weaknesses:**

I am not familiar with this line of research, so I lack the expertise to judge the novelty of this paper, and I apologize for potential misunderstandings in advance.

1. Motivation:
- This paper presents some cool results, but it is still not clear to me why the community would benefit from the polytopal analyses. Furthermore, all analyses are on toy problems that fit closed-form functions.
- One good way to clear up this confusion would be to apply the proposed method on some real-world classifiers (such as ResNet for image classification, or some simple MLPs for various real-world smaller tasks), show that the polytopal analyses reveal properties of the learned neural network that could not be found with existing methods, and discuss how these properties affect real-world applications.

2. Comparison with existing works:
- Line 444 (Humayun et al. (2023) works only for two dimensional inputs) -- the experiments in this paper also considers two dimensions, and Line 134 says "we only investigate curvature only for the two-dimensional case."

3. Other weaknesses:
- Although Figure 8 is nice, it lacks legend and axis labels.
- Typo: Line 150 -- MLP->MPL.
- Typo: Line 234 -- the final "s" in the word "assess" is missing, making it borderline NSFW ;)
- Typo: Line 318 -- the the.

**Questions:**

- Line 235 (checking can further check if any derived vertex lies outside of the input cube) -- what does this mean?
- Why do we need the validity checks in Section 3.2? Does the proposed algorithm not guarantee the validity of its results? If the validity checks fail, what do we do?
- Line 429 (Balestriero & LeCun (2023) has some similarities to our algorithm but differs in scope) -- how is the scope different?

---

> ### Author Response · Authors · 2024-11-25
>
> Dear Reviewer ,
>
> Thank you for reading our paper and commenting on our paper. We revised the paper to address these questions. The main changes are an extend literature review, and novel Section 4.1, we moved the old Section 4.1 to the appendix. In this new section we analyse generalisation properties in particular distributional shifts.
> The algorithm works realistically up to dimension ten, in higher dimension there will be just too many cells. In the paper we considered somewhat hidden in Figure 5, networks of dimension six.
>
> **Questions**
>
> *Line 235 (checking can further check if any derived vertex lies outside of the input cube) -- what does this mean?*
>
> All derived vertices must lie within the bounding cube. If the algorithm outputs a vertex outside of this bounding cube it failed.
>
> *Why do we need the validity checks in Section 3.2? Does the proposed algorithm not guarantee the validity of its results? If the validity checks fail, what do we do?*
>
> As we argue in the paper, the algorithm derives a correct decomposition. What we are assuming in the argument is exact arithmetics. Unfortunately, this is not the case for floating point. We want to use the decomposition in the analysis of a safety critical systems. Thus, we must make sure that our decomposition is correct. The validity checks provide some arguments in this direction. If the validity checks fail further analysis is necessary. Typical errors are missed small cells. If the volume of the found cells covers the input space up to a small error, we might proceed with most parts of the analysis. We provide some further hints, why things can go wrong in the appendix.
>
> *Line 429 (Balestriero & LeCun (2023) has some similarities to our algorithm but differs in scope) -- how is the scope different?*
>
> Balestriero et al. count the number of linear regions, they are not interested the vertices and the substructure which connects these linear regions. The scope of that work is a parallelizable algorithm, and consideration how well sampling based methods perform.

---

### Official Review · Reviewer_BhtF · 2024-11-01

**Soundness:** 3
**Presentation:** 2
**Contribution:** 3
**Rating:** 5
**Confidence:** 3

**Summary:**

This paper analyzes the ReLU-based MLP (piecewise linear activation functions) by viewing their layer representations as polytopes. Based on the analysis, an algorithm is proposed to decompose trained NN into polytope sets and then align them with the training/testing data to assess the performance, which seems to be a single-dimension regression error using MSE. The theoretical analysis specifically focuses on several properties of polytopes, aligning them with the behaviors of NNs. Four typical target functions with different structures and characteristics on polytope separations of input space are used for testing the proposed algorithm. This work is new to the best of the reviewer’s knowledge, but the reviewers have concerns regarding presentation quality and completeness.

**Strengths:**

•	The idea is new in that 1) using a set of polytopes to represent ReLU-based MLP for geometric understanding of layer compositions, and 2) using the polytopes to separate training or testing data points for final outputs.

•	The visualization to align trained NNs with polytope representation is well-understood.

•	The theoretical analysis is not limited to shallow alignment, but translating the properties of polytopes into behaviors of NN layers.

**Weaknesses:**

The reviewer has doubts about the motivation of this work considering the following:

•	What is the purpose of using polytope representation to analyze NNs? For example, the piecewise linear function can also lead to strong convexity.

•	Is the theory only for ReLU-based MLP? While piecewise linear is mentioned, only ReLU-alike activations (e.g., Leaky ReLU) can satisfy this property. If nonlinearity is gradually added, like ELU, is the theory generalizable?


Regarding clarity and completeness of the work:

•	At the beginning of the Introduction, while the example and Figure 1 catch the eye, the explanation is vague, e.g., what is the “symmetry of the data” and what is the difference between the right two plots so that you prefer the right one?

•	“Assess the network” seems to be the target, but it’s unclear what metrics are used to quantify which commonly focused capability of NNs.

•	The algorithm and theoretical analysis mainly discuss the properties of polytopes without sufficient transitions and demonstrations of the NN representation.

•	Four typical target functions are used for testing, each with two inputs. A theoretical analysis may focus on the toy case and intuitive observation, but natural thinking is how researchers can learn or use it for further studies.

**Questions:**

Please refer to the bullet points in Weaknesses.

---

> ### Author Response · Authors · 2024-11-25
>
> Dear reviewer BhtF,
> thank for reading our paper and the raised questions. We revised the paper by writing a new Section 4.1 (moving the old Section 4.1. to the appendix) and updated the Literature review. Below we reply to the points of your review.
>
> • What is the purpose of using polytope representation to analyze NNs? For example, the piecewise linear function can also lead to strong convexity.
>
> In this work we view a NN as a function from $R^D \to R$ with the ultimate goal to perform analysis, such as finding extrema, bounding derivatives and so on. This is motivated as the NN is some surrogate function, hence the need to check its properties. Furthermore, we want to assess how well it fits the data. In 1d the polytopal complex separates an interval in subintervals over each of these intervals the network is affine making it a (continuous) piecewise linear function. In higher dimensions a similar thing is happening. If we want to perform some kind of analysis, we need to do a (local) version of what we described in the paper.
> Most relu feedforward NN lead to piece-wise linear functions. We agree that convexity is a nice property of an NN, but most often it is not the case. Using the polytopal complex one can check if the NN is strongly convex.
>
> • Is the theory only for ReLU-based MLP? While piecewise linear is mentioned, only ReLU-alike activations (e.g., Leaky ReLU) can satisfy this property. If nonlinearity is gradually added, like ELU, is the theory generalizable?
>
> The theory is for all piecewise-linear activation functions, eg. Hardtanh(), abs(). We can also analyse other continuous activation functions, by approximating the nonlinearity by a 1-k-1 ReLU network, this does not increase the complexity to much, as only “parallel” hyperplanes are added to the structure.
> Regarding clarity and completeness of the work:
>
> • At the beginning of the Introduction, while the example and Figure 1 catch the eye, the explanation is vague, e.g., what is the “symmetry of the data” and what is the difference between the right two plots so that you prefer the right one?
>
> Symmetry of the data refers to the symmetry of the circles (the purpose of the network is to classify two concentric circles of different radii). If we would not know this, we still could observe that in the left figure the input domain is separated many cones, whereas in the right figure we only have two regions. The neighbourhood graph of the cells (the linear regions) derived from the polytopal complex would reveal this. Following Occam’s Razor, let’s us choose the right network.
>
> • “Assess the network” seems to be the target, but it’s unclear what metrics are used to quantify which commonly focused capability of NNs.
>
> We made this more clear in Section 4.1 in Tables 1 and 2.
>
> • The algorithm and theoretical analysis mainly discuss the properties of polytopes without sufficient transitions and demonstrations of the NN representation.
>
> We added two new experiments to show how this translates, please see Section 4.1.
>
> • Four typical target functions are used for testing, each with two inputs. A theoretical analysis may focus on the toy case and intuitive observation, but natural thinking is how researchers can learn or use it for further studies.
>
> We think that Section B.1 (old Section 4.1) and Section 4.2 and the new Section 4.1 contain examples how to use the polytopal complex.

---

> > ### Comment · Reviewer_BhtF · 2024-11-27
> >
> > Thanks for the point-to-point answers provided by the authors. My confusions are made clear, especially regarding motivations. I have adjusted the rating, and I recommend the authors include some of the discussions in the paper for clarity.
> >
> > The reviewer has one follow-up comment: "Following Occam’s Razor, let’s us choose the right network." While Occam’s Razor is widely adopted, it may not always be the correct rule, especially for many complex physical systems.

---

### Meta-Review · Area_Chair_83bE · 2024-12-20

**Metareview:**

**Summary**


This paper explores neural networks with ReLU activation function, where the network becomes a continuous piecewise linear entity. The authors introduce an algorithm to decompose the input space into polytopal cells, each behaving as an affine function. This decomposition enables the computation of various network statistics, including maxima, minima, the number of cells, local span, and curvature. This  enables exploration of the network's complexity and efficiency, assessing factors such as cell volume and the impact of network parameters on computational demands. The paper then presents empirical results for several functions, including the Himmelblau and Griewank functions, demonstrating the practical application of the algorithm.

**Strengths**

The reviewers unanimously highlighted several strengths of the proposed framework:

* The visualizations in the paper are particularly helpful for understanding the contributions of the paper.
* The  proposed algorithm for computing the polytopal complex of neural networks is novel and interesting, and it enables  detailed analysis of properties like maxima, minima, and curvature, and exploring the impact of network parameters such as depth, width, and regularization.
* Various aspects of the paper including  the analysis of the decomposition time,  the validation of the polyhedral complex, and the analysis on the effect of regularization on the polyhedral complex were praised by the reviewers.



**Weaknesses**

Several core weaknesses was brought up by the reviewers. These include:
* The presentation of the paper could be improved, as evidenced by the numerous confusions expressed by the reviewers.
*  The time complexity of the algorithm in this paper is very high.
* The related work section of the paper missed very relevant work in the literature, which leads to a poor placement of the paper in the literature. In addition, and in light of the existing work in the literature, the novelty of the paper seems to be very marginal.

**Conclusion**

The majority of reviewers found the paper's approach interesting, but noted significant misunderstandings and confusions, indicating that certain aspects of the paper are convoluted and not clearly explained. Moreover, there were concerns about the practical implications of the paper. Despite the authors' rebuttal, the prevailing negative perception among reviewers remained unchanged. I agree that the paper is not ready for publication in its current form and vote to reject it.

**Additional Comments On Reviewer Discussion:**

Given the less polarized evaluations of this paper, the majority of reviewers found the paper not ready for publication, which I concur with.

---

### Decision · Program_Chairs · 2025-01-22

Reject